Assessment and economic valuation of air pollution impacts on human health over Europe
and the United States as calculated by a multi-model ensemble in the framework of
AQMEII3
Ulas Im[1*], Jørgen Brandt[1], Camilla Geels[1], Kaj Mantzius Hansen[1], Jesper Heile Christensen[1],
Mikael Skou Andersen[1], Efisio Solazzo[2], Ioannis Kioutsioukis[3], Ummugulsum Alyuz[4],
Alessandra Balzarini[5], Rocio Baro[6], Roberto Bellasio[7], Roberto Bianconi[7], Johannes Bieser[8],
Augustin Colette[9], Gabriele Curci[10,11], Aidan Farrow[12], Johannes Flemming[13], Andrea
Fraser[14], Pedro Jimenez-Guerrero[6], Nutthida Kitwiroon[15], Ciao-Kai Liang[16], Uarporn
Nopmongcol[17], Guido Pirovano[5], Luca Pozzoli[4,2], Marje Prank[18,19], Rebecca Rose[14], Ranjeet
Sokhi[12], Paolo Tuccella[10,11], Alper Unal[4], Marta Garcia Vivanco[9,20], Jason West[16], Greg
Yarwood[17], Christian Hogrefe[21], Stefano Galmarini[2]
[1] Aarhus University, Department of Environmental Science, Frederiksborgvej 399, DK-4000,
Roskilde, Denmark.
[2] European Commission, Joint Research Centre (JRC), Ispra (VA), Italy.
[3] University of Patras, Department of Physics, University Campus 26504 Rio, Patras, Greece
[4] Eurasia Institute of Earth Sciences, Istanbul Technical University, Istanbul, Turkey
[5] Ricerca sul Sistema Energetico (RSE SpA), Milan, Italy
[6] University of Murcia, Department of Physics, Physics of the Earth, Campus de Espinardo, Ed.
CIOyN, 30100 Murcia, Spain
[7] Enviroware srl, Concorezzo, MB, Italy
[8] Institute of Coastal Research, Chemistry Transport Modelling Group, Helmholtz-Zentrum
Geesthacht, Germany
[9] INERIS, Institut National de l'Environnement Industriel et des Risques, Parc Alata, 60550 Verneuil-
en-Halatte, France
[10] Dept. Physical and Chemical Sciences, University of L'Aquila, L'Aquila, Italy
[11] Center of Excellence CETEMPS, University of L'Aquila, L'Aquila, Italy
[12] Centre for Atmospheric and Instrumentation Research (CAIR), University of Hertfordshire,
Hatfield, UK
[13] European Centre for Medium Range Weather Forecast (ECMWF), Reading, UK
[14] Ricardo Energy & Environment, Gemini Building, Fermi Avenue, Harwell, Oxon, OX11 0QR, UK
[15] Environmental Research Group, Kings' College London, London, UK
[16] Department of Environmental Sciences and Engineering, University of North Carolina at Chapel
Hill, Chapel Hill, North Carolina, USA
[17] Ramboll Environ, 773 San Marin Drive, Suite 2115, Novato, CA 94998, USA
[18] Finnish Meteorological Institute, Atmospheric Composition Research Unit, Helsinki, Finland
[19] Cornell University, Department of Earth and Atmospheric Sciences, Ithaca, USA
[20] CIEMAT. Avda. Complutense 40., 28040 Madrid, Spain
[21] Computational Exposure Division, National Exposure Research Laboratory, Office of Research and
Development, United States Environmental Protection Agency, Research Triangle Park, NC, USA.
*Correspondence to:* Ulas Im (ulas@envs.au.dk)
**Abstract**
The impact of air pollution on human health and the associated external costs in Europe and
the United States (U.S.) for the year 2010 is modelled by a multi-model ensemble of regional
models in the frame of the third phase of the Air Quality Modelling Evaluation International
Initiative (AQMEII3). The modelled surface concentrations of $O_3$, $CO$, $SO_2$ and $PM_{2.5}$ are
used as input to the Economic Valuation of Air Pollution (EVA) system to calculate the
resulting health impacts and the associated external costs from each individual model. Along
with a base case simulation, additional runs were performed introducing 20% anthropogenic
emission reductions both globally and regionally in Europe, North America and East Asia, as
defined by the second phase of the Task Force on Hemispheric Transport of Air Pollution
(TF-HTAP2).
Health impacts estimated by using concentration inputs from different chemistry and
transport models (CTMs) to the EVA system can vary up to a factor of three in Europe
(twelve models) and the United States (three models). In Europe, the multi-model mean total
number of premature deaths (acute + chronic) is calculated to be 414 000 while in the U.S., it
is estimated to be 160 000, in agreement with previous global and regional studies. The
economic valuation of these health impacts are calculated to be 300 and 145 billion Euros in
Europe and the U.S., respectively.  A subset of models that produce the smallest error
compared to the surface observations at each time step against an all-models mean ensemble
results in increase of health impacts by up to 30% in Europe, while in the U.S., the optimal
ensemble mean led to a decrease in the calculated health impacts by ~11%.
A total of 54 000 and 27 500 premature deaths can be avoided by a 20% reduction of global
anthropogenic emissions in Europe and the U.S., respectively. A 20% reduction of North
American anthropogenic emissions avoids a total premature death of ~1 000 in Europe and
25 000 total premature deaths in the U.S. A 20% decrease of anthropogenic emissions within
the European source region avoids a total premature death of 47 000 in Europe. Reducing the
East Asian anthropogenic emissions by 20% avoids ~2000 total premature deaths in the U.S.
These results show that the domestic anthropogenic emissions make the largest impacts on
premature death on a continental scale, while foreign sources make a minor contributing to
adverse impacts of air pollution.
**1. Introduction**
According to the World Health Organization (WHO), air pollution is now the world's largest
single environmental health risk (WHO, 2014). Around 7 million people died prematurely in
2012 as a result of air pollution exposure from both outdoor and indoor emission sources
(WHO, 2014). WHO estimates 3.7 million premature deaths in 2012 from exposure to
outdoor air pollution from urban and rural sources worldwide. According to the Global
Burden of Disease (GBD) study, exposure to ambient particulate matter pollution remains
among the ten leading risk factors. Air pollution is a transboundary phenomenon with global,
regional, national and local sources, leading to large differences in the geographical
distribution of human exposure. Short-term exposure to ozone ($O_3$) is associated with
respiratory morbidity and mortality (e.g. Bell et al., 2004), while long-term exposure to $O_3$
has been associated with premature respiratory mortality (Jerrett et al., 2009). Short-term
exposure to particulate matter ($PM_{2.5}$) has been associated with increases in daily mortality
rates from respiratory and cardiovascular causes (e.g. Pope and Dockery, 2006), while long-
term exposure to $PM_{2.5}$ can have detrimental chronic health effects, including premature
mortality due to cardiopulmonary diseases and lung cancer (Burnett et al., 2014). The Global
Burden of Disease Study 2015 estimated 254 000 $O_3$-related and 4.2 million anthropogenic
$PM_{2.5}$-related premature deaths per year (Cohen et al., 2017).
Changes in emissions from one region can impact air quality over others, affecting also air
pollution-related health impacts due to intercontinental transport (Anenberg et al., 2014;
Zhang et al., 2017). In the framework of the Task Force on Hemispheric Transport of Air
Pollution (TF-HTAP), Anenberg et al. (2009) found that reduction of foreign ozone precursor
emissions can contribute to more than 50% of the deaths avoided by simultaneously reducing
both domestic and foreign precursor emissions. Similarly, they found that reducing emissions
in North America (NA) and Europe (EU) has largest impacts on ozone-related premature
deaths in downwind regions than within (Anenberg et al., 2009). This result agrees with
Duncan et al. (2008), which showed for the first time that emission reductions in NA and EU
have greater impacts on ozone mortality outside the source region than within. Anenberg et
al. (2014) estimates that 93–97 % of $PM_{2.5}$-related avoided deaths from reducing emissions
occurs within the source region while 3–7 % occur outside the source region from
concentrations transported between continents. In spite of the shorter lifetime of $PM_{2.5}$
compared to $O_3$, it was found to cause more deaths from intercontinental transport (Anenberg
et al., 2009; 2014). In the frame of the second phase of the Task Force on Hemispheric
Transport of Air Pollution (TF-HTAP2; Galmarini et al., 2017), an ensemble of global
chemical transport model simulations calculated that 20% emission reductions from one
region generally lead to more avoided deaths within the source region than outside (Liang et
al., 2017).
Recently, Lelieveld et al. (2015) used a global chemistry model and calculated that outdoor
air pollution led to 3.3 million premature deaths globally in 2010. They calculated that in
Europe and North America, 381 000 and 68 000 premature deaths occurred, respectively.
They have also calculated that these numbers are likely to roughly double in the year 2050
assuming a business-as-usual scenario. Silva et al. (2016), using the ACCMIP model
ensemble, calculated that the global mortality burden of ozone is estimated to markedly
increase from 382 000 deaths in 2000 to between 1.09 and 2.36 million in 2100. They also
calculated that the global mortality burden of $PM_{2.5}$ is estimated to decrease from 1.70
million deaths in 2000 to between 0.95 and 1.55 million deaths in 2100. Silva et al. (2013)
estimated that in 2000, 470 000 premature respiratory deaths are associated globally and
annually with anthropogenic ozone, and 2.1 million deaths with anthropogenic $PM_{2.5}$-related
cardiopulmonary diseases (93%) and lung cancer (7%). These studies employed global
chemistry and transport models with coarse spatial resolution ($\geq 0.5°\times0.5°$), therefore health
benefits from reducing local emissions were not able to be adequately captured. Higher
resolutions are necessary to calculate more robust estimates of health benefits from local vs.
non-local sources (Fenech et al., 2017). In addition, these studies calculated number of
premature deaths due to air pollution, however none of them addresses morbidity such as
number of lung cancer or asthma cases, or restricted activity days. Finally, these studies did
not include economic costs either. On the other hand, there are a number of regional studies
that calculate health impacts on finer spatial resolutions, and address morbidity. However,
they are mostly based on single air pollution models or do not evaluate the health benefits
from local vs. non-local emissions. Therefore, a comprehensive study employing multi model
ensemble of high spatial resolution and focusing on both mortality and morbidity from local
vs. non-local sources lacks in the literature.
In Europe, recent results show that outdoor air pollution due to $O_3$, $CO$, $SO_2$ and $PM_{2.5}$ causes
a total number of 570 000 premature deaths in the year 2011 (Brandt et al., 2013a; 2013b).
The external (or indirect) costs to society related to health impacts from air pollution are
tremendous. OECD (2014) estimates that outdoor air pollution is costing its member
countries USD 1.57 trillion in 2010. Among the OECD member countries, the economic
valuation of air pollution in the U.S. was calculated to be ~500 billion USD and ~660 USD in
Europe. In the whole of Europe, the total external costs have been estimated to approx. 800
billion Euros in year 2011 (Brandt et al., 2013a). These societal costs have great influence on
the general level of welfare and especially on the distribution of welfare both within the
countries as air pollution levels are vastly heterogeneous both at regional and local scales and
between the countries as air pollution and the related health impacts are subject to long-range
transport. Geels et al. (2015), using two regional chemistry and transport models, estimated a
premature mortality of 455 000 and 320 000 in Europe (EU28 countries) for the year 2000,
respectively, due to $O_3$, $CO$, $SO_2$ and $PM_{2.5}$. They also estimated that climate change alone
leads to a small increase (15%) in the total number of $O_3$-related acute premature deaths in
Europe towards the 2080s and relatively small changes (<5%) for $PM_{2.5}$-related mortality.
They found that the combined effect of climate change and emission reductions will reduce
the premature mortality due to air pollution, in agreement with the results from Schucht et al.
151   (2015).

The U.S. Environmental Protection Agency estimated that in 2010 there were ~160 000
premature deaths in the U.S. due to air pollution (U.S. EPA, 2011). Fann et al. (2012)
calculated 130,000 - 350,000 premature deaths associated with $O_3$ and $PM_{2.5}$ from the
anthropogenic sources in the U.S. for the year 2005. Caizzo et al. (2013) estimated 200 000
cases of premature death in the U.S. due to air pollution from combustion sources for the year
157   2005.

The health impacts of air pollution and their economic valuation are estimated based on
observed and/or modelled air pollutant concentrations. Observations have spatial limitations
particularly when assessments are needed for large regions. The impacts of air pollution on
health can be estimated using models, where the level of complexity can vary depending on
the geographical scale (global, continental, country or city), concentration input
(observations, model calculations, emissions) and the pollutants of interest that can vary from
only few ($PM_{2.5}$ or $O_3$) to a whole set of all regulated pollutants. The health impact models
normally used may differ in the geographical coverage, spatial resolutions of the air pollution
model applied, complexity of described processes, the exposure-response functions (ERFs),
population distributions and the baseline indices (see Anenberg et al., 2015 for a review).
Air pollution related health impacts and associated costs can be calculated using Chemical
Transport Model (CTM) or with standardized source-receptor relationships characterizing the
dependence of ambient concentrations on emissions. (e.g. EcoSense model: ExternE, 2005,
TM5-FASST: Van Dingenen et al., 2014). Source-receptor relationships have the advantage
of reducing the computing time significantly and have therefore been extensively used in
systems like GAINS (Amann et al., 2011). On the other hand, full CTM simulations have the
advantage of better accounting for non-linear chemistry-transport processes in the
atmosphere.
CTMs are useful tools to calculate the concentrations of health-related pollutants taking into
account non-linearities in the chemistry and the complex interactions between meteorology
and chemistry. However, the CTMs include different chemical and aerosol schemes that
introduce differences in the representation of the atmosphere as well as differences in the
emissions and boundary conditions they use (Im et al., 2015a,b). These different approaches
are present also in the health impact estimates that use CTM results as basis for their
calculations. Multi-model  (MM) ensembles can be useful to the extent that allow us to take
into consideration several model results at the same time, define the relative weight of the
various members in determining the mean behavior, and  produce also an uncertainty
estimated based on the diversity of the results (Potempski and Galmarini, 2010; Riccio et al.,
2013;  Solazzo et al., 2013).
The third phase of the Air Quality Modelling Evaluation International Initiative (AQMEII3)
project brought together fourteen European and North American modelling groups to
simulate the air pollution levels over the two continental areas for the year 2010 (Galmarini et
al., 2017). Within AQMEII3, the simulated surface concentrations of health related air
pollutants from each modelling group serves as input to the Economic Valuation of Air
Pollution (EVA) model (Brandt et al., 2013a; 2013b). The EVA model is used to calculate the
impacts of health-related pollutants on human health over the two continents as well as the
associated external costs. EVA model has also been tested and validated for the first time
outside Europe. We adopt a multi-model ensemble (MM) approach, in which the outputs of
the modelling systems are statistically combined assuming equal contribution from each
model and used as input for the EVA model. In addition, the human health impacts (and the
associated costs) of reducing anthropogenic emissions, globally and regionally have been
calculated, allowing to quantify the trans-boundary benefits of emission reduction strategies.
Finally, following the conclusions of Solazzo and Galmarini (2015), the health impacts have
been calculated using an optimal ensemble of models, determined by error minimization .
This approach can assess the health impacts with reduced model bias, which we can then
compare with the classically derived estimates based on model averaging.
**2. Material and Methods**
**2.1. AQMEII**
*2.1.1. Participating Models*
In the framework of the AQMEII3 project, fourteen groups participated to simulate the air
pollution levels in Europe and North America for the year 2010. In the present study, we use
results from the thirteen groups that provided all health-related species (Table 1). As seen in
Table 1, six groups have operated the CMAQ model. The main differences among the CMAQ
runs reside in the number of vertical levels and horizontal spacing (Table 1) and in the
estimation of biogenic emissions. UK1, DE1, and US3 calculated biogenic emissions using the
BEIS (Biogenic Emission Inventory System version 3) model, while TR1, UK1, and UK2
calculated biogenic emissions through the MEGAN model (Guenther et al., 2012). Moreover,
DE1 does not include the dust module, while the other CMAQ instances use the inline
calculation (Appel et al., 2013) and TR1 uses the dust calculation previously calculated for
AQMEII Phase 2. Finally, all runs were carried out using CMAQ version 5.0.2 except for TR1,
which is based on the 4.7.1 version. The gas-phase mechanisms and the aerosol models are
used by each group is also presented in Table 1.More details of the model system are provided
in the supplementary material. The differences in the meteorological drivers and aerosol
modules can lead to substantial differences in modelled concentrations (Im et al., 2015b).
*2.1.2. Emission and Boundary Conditions*
The base-case emission inventories that are used in AQMEII for Europe and North America
are extensively described in Pouliot et al. (2015). For Europe, the 2009 inventory of TNO-
MACC anthropogenic emissions was used. In regions not covered by the emission inventory,
such as North Africa, five modelling systems have complemented the standard inventory with
the HTAPv2.2 datasets (Janssens-Maenhout et al., 2015). For the North American domain,
the 2008 National Emission Inventory was used as the basis for the 2010 emissions,
providing the inputs and datasets for processing with the SMOKE emissions processing
system (Mason et al., 2012). For both continents the regional scale emission inventories were
embedded in the global scale inventory (Janssens-Maenhout et al., 2015) used by the global-
scale HTAP2 modelling community so that to guarantee coherence and harmonization of the
information used by the regional scale modelling community. The annual totals for European
and North American emissions in the HTAP inventory are the same as the MACC and
SMOKE emissions. However, there are differences in the temporal distribution, chemical
speciation and the vertical distribution used in the models. The C-IFS model (Flemming et
al., 2015 and 2017) provided chemical boundary conditions. The C-IFS model has been
extensively evaluated in Flemming et al. (2015 and 2017), and in particular for North
America (Hogrefe et al., 2017; Huang et al., 2017). Galmarini et al. (2017) provides more
details on the setup of the AQMEII3 and HTAP2 projects.
*2.1.3. Model Evaluation*
The models' performance on simulating the surface concentrations of the health-related
pollutants were evaluated using Pearson's Correlation (*r*), normalized mean bias (*NMB*),
normalized mean gross error (*NMGE*) and root mean square error (*RMSE*) to compare the
modelled and observed hourly pollutant concentrations over surface measurement stations in
the simulation domains. The hourly modelled vs. observed pairs are averaged and compared
on a monthly basis. The modelled hourly concentrations were first filtered based on
observation availability before the averaging has been performed. The observational data
used in this study are the same as the dataset used in second phase of AQMEII (Im et al.,
2015a, b). Surface observations are provided in the Ensmeble system
(http://ensemble2.jrc.ec.europa.eu/public/) that is hosted at the Joint Research Centre (JRC).

Observational data were originally derived from the surface air quality monitoring networks operating in EU and NA. In EU, surface data were provided by the European Monitoring and Evaluation Programme (EMEP, 2003; http://www.emep.int/) and the European Air Quality Database (AirBase; http://acm.eionet.europa.eu/databases/airbase/). In NA observational data were obtained from the NAtChem (Canadian National Atmospheric Chemistry) database and from the Analysis Facility operated by Environment Canada (http://www.ec.gc.ca/natchem/).

The model evaluation has been conducted for 491 European and 626 North American stations for $O_3$, 541 European stations and 37 North American stations for CO, 500 European station and 277 North American stations for $SO_2$, and 568 European stations and 156 North American stations for $PM_{2.5}$.

*2.1.4. Emissions Perturbations*

In addition to the base case simulations in AQMEII3, a number of emission perturbation scenarios have been simulated (Table 1). The perturbation scenarios feature a reduction of 20% in the global anthropogenic emissions (GLO) as well as the HTAP2-defined regions of Europe (EUR), North America (NAM) and East Asia (EAS), as explained in detail in Galmarini et al. (2017) and Im et al. (2017). To prepare these scenarios, both the regional models and the global C-IFS model that provides the boundary conditions to the participating regional models have been operated with the reduced emissions. The global perturbation scenario (GLO) reduces the global anthropogenic emissions by 20%, introducing a change in the boundary conditions as well as a 20% decrease in the anthropogenic emissions used by the regional models. The North American perturbation scenario (NAM) reduces the anthropogenic emissions in North America by 20%, introducing a change in the boundary conditions while anthropogenic emissions remain unchanged for Europe, showing the impact of long-range transport while for North America, while the scenarios introduces a 20% reduction of anthropogenic emissions in the HTAP-defined North American region. The European perturbation scenario (EUR) reduces the anthropogenic emissions in the HTAP-defined Europe domain by 20%, introducing a change in the anthropogenic emissions while boundary conditions remain unchanged in the regional models, showing the contribution from the domestic anthropogenic emissions only. Finally, the East Asian perturbation scenario (EAS) reduces the anthropogenic emissions in East Asia by 20%, introducing a change in the boundary conditions while anthropogenic emissions remain unchanged in the regional models, showing the impact of long-range transport from East Asia on the NA concentrations.

**2.2. Health Impact Assessment**

All modeling groups interpolate their model outputs on a common 0.25°×0.25° resolution AQMEII grid predefined for Europe (30°W - 60°E, 25°N - 70°N) and North America (130°W - 59.5°W, 23.5°N - 58.5°N). All the analyses performed in the present study use the pollutant concentrations on these final grids. Health impacts are first calculated for each individual model and then the ensemble mean, median and standard deviation are calculated

for each health impact. In order to be able to estimate an uncertainty in the health impacts calculations, none of the models were removed from the ensemble.

Along with the individual health impact estimates from each model, a multi-model mean dataset ($MM_m$, in which all the modelling systems are averaged assuming equally weighted contributions) has been created for each grid cell and time step, hence creating a new model set of results that have the same spatial and temporal resolution of the ensemble-contributing members. In addition to this simple $MM_m$, an optimal MM ensemble ($MM_{opt}$) has been generated. $MM_{opt}$ is created following the criteria extensively discussed and tested in the previous phases of the AQMEII activity (Riccio et al., 2012; Kioutsioukis et al., 2016; Solazzo and Galmarini, 2016), where it was shown that there are several ways to combine the ensemble members to obtain a superior model, mostly depending on the feature we wish to promote (or penalize). For instance, generating an optimal ensemble that maximizes the accuracy would require a minimization of the mean error or of the bias, while maximizing the associativity (variability) would require maximize the correlation coefficient (standard deviation). In this study, the sub-set of models whose mean minimize the mean squared error ($MSE$) is selected as optimal ($MM_{opt}$). $MM_m$ and $MM_{opt}$ have therefore the same spatial resolution with the individual models. The $MSE$ is chosen for continuity with previous AQMEII-related works. The $MSE$ is chosen in the light of its property of being composed by bias, variance and covariance types of error, thus lumping together measures of accuracy (bias), variability (variance) and associativity (covariance) (Solazzo and Galmarini, 2016). The minimum $MSE$ has been calculated at the monitoring stations, where observational data are available and then extended to the entire continental areas. This approximation might affect remote regions away from the measurements. However, considering that for the main pollutants ($O_3$ and $PM_{2.5}$) the network of measurements is quite dense around densely populated areas (where the inputs of the MM ensemble are used for assessing the impact of air pollutants on the health of the population), errors due to inaccurate model selection in remote regions might be regarded as negligible (Solazzo and Galmarini, 2015). It should be noted that the selection of the optimal combinations of models is affected by the model's bias that might stem from processes that are common to all members of the ensemble (e.g. emissions). Therefore, such a common bias does not cancel out when combining the models, possibly creating a biased ensemble. Current work is being devoted to identify the optimal combinations of models from which the offsetting bias is removed (Solazzo et al., 2017b).

### *2.2.1. EVA System*

The EVA system (Brandt et al., 2013a, b) is based on the impact-pathway chain (e.g. Friedrich and Bickel, 2001), consisting of the emissions, transport and chemical transformation of air pollutants, population exposure, health impacts and the associated external costs. The EVA system requires hourly gridded concentration input from a regional-scale CTM as well as gridded population data, exposure-response functions (ERFs) for health impacts, and economic valuations of the impacts from air pollution. A detailed description of the integrated EVA model system along with the ERFs and the economic valuations used are given in Brandt et al. (2013a).

The gridded population density data over Europe and the U.S. used in this study are presented in Fig. 1. The population data over Europe are provided on a 1km spatial resolution from Eurostat for the year 2011 (http://www.efgs.info).  The U.S. population data has been provided from the U.S. Census Bureau for the year 2010. The total populations used in this study are roughly 532 and 307 million in Europe and the U.S., respectively. As the health outcomes are age-dependent, the total population data has been broken down to a set of age intervals being babies (under 9 months), children (under 15), adult (above 15), above 30, and above 65. The fractions of population in these intervals for Europe is derived from the EUROSTAT 2000 database, where the number of persons of each age at each grid cell was aggregated into the above clusters (Brandt et al., 2011), while for the U.S. they are derived from the U.S. Census Bureau for the year 2010 at 5-year intervals.

The EVA system can be used to assess the number of various health outcomes including different morbidity outcomes as well as short-term (acute) and long-term (chronic) mortality, related to exposure of $O_3$, CO and $SO_2$ (short-term) and $PM_{2.5}$ (long-term). Furthermore, impact on infant mortality in response to exposure of $PM_{2.5}$ is calculated. The health impacts are calculated using an ERF of the following form:

$$R = \alpha \times \ \delta_c \times P$$

where $R$ is the response (in cases, days, or episodes), $c$ denotes the pollutant concentration, $P$ denotes the affected share of the population, and $\alpha$ an empirically determined constant for the particular health outcome. EVA uses ERFs that are modelled as a linear function, which is a reasonable approximation as showed in several studies (e.g. Pope et al., 2000; the joint World Health Organization/UNECE Task Force on Health (EU, 2004; Watkiss et al., 2005)). Many epidemiological studies have analyzed the concentration-response relationship between ambient PM and mortality using various statistical models. In general, the shapes of the estimated curves did not differ significantly from linear. However, some studies showed non-linear relationships, being steeper at lower than at higher concentrations (e.g. Samoli et al., 2005). Therefore, linear relationships may lead to overestimated health impacts over highly polluted areas. The concentration metrics used in each ERF is shown in Table 2. The sensitivity of EVA to the different pollutant concentrations are further evaluated in the supplementary material and depicted in Fig. S1. EVA calculates and uses the annual mean concentrations of CO, $SO_2$ and $PM_{2.5}$, while for $O_3$, it uses the SOMO35 metric that is defined as the yearly sum of the daily maximum of 8-hour running average over 35 ppb, following WHO (2013) and EEA (2017).

The morbidity outcomes include chronic bronchitis, restricted activity days, congestive heart failure, lung cancer, respiratory and cerebrovascular hospital admissions, asthmatic children (<15 years) and adults (>15 years), which includes bronchodilator use, cough, and lower respiratory symptoms. The exposure-response functions are broadly in line with estimates derived with detailed analysis in EU funded research (Rabl, Spadaro and Holland, 2014; EEA, 2013) To figure out the total number of premature deaths from the years of life lost due to $PM_{2.5}$,  they have been converted into lost lives according to a lifetable method (explained in detail in Andersen, 2017) but using the factor of 10.6, as reported by (Watkiss et al., 2005).

To these deaths are added the acute deaths due to $O_3$ and $SO_2$. The ERFs used, along with
their references, in both continents as well as the economic valuations for each health
outcome in Europe and the U.S., respectively, are presented in Table 2. Baseline incidence
rates are not assumed to be dissimilar, which is a coarse approach for morbidity. The baseline
rates are from Statistics Denmark
(http://www.statistikbanken.dk/statbank5a/default.asp?w=1280) and lifetables are based on
Denmark, which is close to the US and Eurozone average (Andersen, 2017). For a description
of the morbidity ERFs, see Andersen et al. (2004 and 2008). The economic valuations are
provided by Brandt et al. (2013a); see also EEA (2013).
ERF for all-cause chronic mortality due to $PM_{2.5}$ were based on the findings of Pope et al.
(2002), which is the most extensive study available, following conclusions from the scientific
review of the Clean Air For Europe (CAFÉ) programme (Hurley et al., 2005; Krupnick et al.,
2005). The results from Pope et al. (2002) are further supported by Krewski et al. (2009), and
more recently by the latest HRAPIE project report (WHO, 2013a). Therefore, as
recommended by WHO (2013a), EVA uses the ERFs based on the meta-analysis of 13 cohort
studies as described in Hoek et al. (2013). In EVA, the number of lost life years for a Danish
population cohort with normal age distribution, when applying the ERF of Pope et al. (2002)
for all-cause mortality (relative risk, RR= 1.062 (1.040-1.083) on 95% confidence interval),
and the latency period indicated, sums to 1138 yr of life lost (YOLL) per 100 000 individuals
for an annual $PM_{2.5}$ increase of 10 μg m$^{-3}$ (Andersen, 2008)..EVA uses a counterfactual
$PM_{2.5}$ concentration of 0 μgm$^{-3}$ following the EEA methodology, meaning that the impacts
have been estimated for the full range of modelled concentrations from 0 μgm$^{-3}$ upwards.
Applying a low counterfactual concentration can underestimate health impacts at low
concentrations if the relationship is linear or close to linear (Anenberg et al., 2016). However,
it is important to note that uncertainty in the health impact results may increase at low
concentrations due to sparse epidemiological data. Assuming linearity at very low
concentrations may distort the true health impacts of air pollution in relatively clean
atmospheres (Anenberg et al., 2016).
It has been shown that $O_3$ concentrations above the level of 35 ppb involve an acute mortality
increase, presumably for weaker and elderly individuals. EVA applies the ERFs selected in
CAFE for post-natal death (age group 1–12 months) and acute death related to $O_3$ (Hurley et
al., 2005). WHO (2013a) also recommends the use of the daily maximum of 8-hour mean $O_3$
concentrations for the calculation of the acute mortality due to $O_3$. There are also studies
showing that $SO_2$ is associated with acute mortality, and EVA adopts the ERF identified in
the APHENA study – Air Pollution and Health: A European Approach (Katsouyanni et al.,
408 1997).

Chronic exposure to $PM_{2.5}$ is also associated with morbidity, such as lung cancer. EVA
employs the specific ERF (RR = 1.08 per 10 μg m$^{-3}$ $PM_{2.5}$ increase) for lung cancer indicated
in Pope et al. (2002). Bronchitis has been shown to increase with chronic exposure to $PM_{2.5}$
and we apply an ERF (RR = 1.007) for new cases of bronchitis based on the AHSMOG study
(involving non-smoking Seventh-Day Adventists; Abbey et al., 1999), which is the same
epidemiological study as in CAFE (Abbey, 1995; Hurley et al., 2005). The ExternE crude

incidence rate was chosen as a background rate (ExternE, 1999), which is in agreement with a Norwegian study, rather than the pan-European estimates used in CAFE (Eagan et al., 2002). Restricted activity days (RADs) comprise two types of responses to exposure: so-called minor restricted activity days as well as work-loss days (Ostro, 1987). This distinction enables accounting for the different costs associated with days of reduced well-being and actual sick days. It is assumed that 40% of RADs are work-loss days based on Ostro (1987). The background rate and incidence are derived from ExternE (1999). Hospital admissions are deducted to avoid any double counting. Hospital admissions and health effects for asthmatics (here corresponding to the three responses bronchodilator use, cough and lower respiratory symptoms) are also based on ExternE (1999).

Table 2 lists the specific valuation estimates applied in the modelling of the economic valuation of mortality and morbidity effects. A principal value of EUR 1.5 million was applied for preventing an acute death, following expert panel advice (EC 2001). For the valuation of a life year, the results from a survey relating specifically to air pollution risk reductions were applied (Alberini et al., 2006), implying a value of EUR 57.500 per year of life lost (YOLL). With the more conservative metric of estimating lost life years, rather than 'full' statistical lives, there is no adjustment for age. This is due to the fact that government agencies in Europe, including the European Commission, apply a methodology for costing of air pollution that is based on accounting for lost life years, rather than for entire statistical lives as is customary in USA. While the average traffic victim, for instance, is mid-aged and likely to lose about 35-40 years of life expectancy, pollution victims are believed to suffer significantly smaller losses of years (EAHEAP, 1999:64; Friedrich and Bickel, 2001). To avoid overstating the benefits of air pollution control, these are treated as proportional to the number of life years lost. Most of the excess mortality is due to chronic exposure to air pollution over many years and the life year metric is based on the number of lost life years in a statistical cohort. Following the guidelines of the Organisation for Economic Co-operation and Development (OECD, 2006), the predicted acute deaths, mainly from $O_3$, are valuated here with the adjusted value for preventing a fatality (VSL, Value of a Statistical Life). The life tables are obtained from European data and are applied to the U.S. as the average life expectancy in the U.S. is similar to that in Europe, and close to the OECD average (OECD, 2016). The willingness to pay for reductions in risk obviously differs across income levels. However, in the case of air pollution costs, adjustment according to per capita income differences among different states is not regarded as appropriate, because long-range transport implies that emissions from one state will affect numerous other states and their citizens. The valuations are thus adjusted with regional purchasing power parities (PPP) of EU27 and USA.

Cost-benefit analysis in the U.S. related to air pollution proceeds from a standard approach, where abatement measures preventing premature mortality are considered according to the number of statistical fatalities avoided, which are appreciated according to the value of VSL (presently USD 7.4 million). In contrast, and following recommendations from the UK working group on Economic Appraisal of the Health Effects of Air Pollution (EAHEAP, 1999), focus in EU has been on the possible changes in average life expectancy resulting

from air pollution. In EU, the specific number of life years lost as a result of changes in air
pollution exposures are estimated based on lifetable methodology, and monetized with Value-
Of-Life-Year (VOLY) unit estimates (Holland et al. 1999; Leksell and Rabl 2001). The
theoretical basis is a life-time consumption model according to which the preferences for risk
reduction will reflect expected utility of consumption for remaining life years (Hammitt
2007; OECD 2006:204). The much lower VSL values customary in Europe (presently €2.2
million) add decisively to the differences, as VOLY is deducted from this value. By using a
common valuation framework according the EU approach we allow for direct comparisons of
the monetary results. It follows from OECD recommendations (2012) to correct with PPP
when doing such benefit transfer. The unit values have been indexed to 2013 prices as
indicated in Table 2.

**3. Results**
3.1. Model Evaluation
Observed and simulated hourly surface $O_3$, CO, $SO_2$ and daily $PM_{2.5}$, which are species used
in the EVA model to calculate the health impacts, over Europe and North America for the
entire 2010 were compared in order to evaluate each model's performance. The statistical
parameters to evaluate the models and their equations are provided in the supplementary
material. For a more thorough evaluation of models and species, see Solazzo et al. (2017a).
The results of this comparison are presented in Table S1 for EU and NA, along with the
multi-model mean and median values. The monthly time series plots of observed and
simulated health-related pollutants are also presented in Figs. 2 and 3. The monthly means are
calculated using the hourly pairs of observed and modelled concentrations at each station.
The results show that over Europe, the temporal variability of all gaseous pollutants is well
captured by all models with correlation coefficients (*r*) higher than 0.70 in general. The
normalized mean biases (*NMB*) in simulated $O_3$ levels are generally below 10% with few
exceptions up to -35%. CO levels are underestimated by up to 45%, while the majority of the
models underestimated $SO_2$ levels by up to 68%, while some models overestimated $SO_2$ by
up to 49%. $PM_{2.5}$ levels are underestimated by 19% to 63%. Over Europe, the median of the
ensemble performs better than the mean in terms of model bias (*NMB*) for $O_3$ (by 52%),
while for CO, $SO_2$ and $PM_{2.5}$, the mean performs slightly better than the median (Table S1).
We have further evaluated the models' performance on simulating the annual mean pollutant
levels over individual measurements stations and plotted the geographical distribution of the
bias. Fig. 4 presents the multi model mean geographical distribution of bias from daily max
8-hour (DM8H) average $O_3$, CO, $SO_2$ and $PM_{2.5}$ over Europe, while Fig. S2-S5 show annual
mean bias for $O_3$, CO, $SO_2$ and $PM_{2.5}$ for each model, respectively. DM8H $O_3$ levels over
Europe are generally underestimated by up to 50 $\mu gm^{-3}$, with few overestimations up to 50
$\mu gm^{-3}$ over southern Europe (Fig. 4a)  The geographical pattern of annual mean $O_3$ bias is
similar among the models with slight differences ($\pm$ 10 $\mu gm^{-3}$) in the bias (Fig. S2). CO
levels are underestimated over all stations by up to 600 $\mu gm^{-3}$ except for few stations where
CO levels are overestimated by up to 100 $\mu gm^{-3}$ (Fig. 4b). All models underestimated CO
levels over the majority of the stations (Fig. S3). $SO_2$ levels are slightly overestimated over
central and southern Europe (Fig. 4c). There are also underestimation over few stations with
no specific geographical pattern. Similar to CO, all models underestimated SO2 levels over
the majority of the stations (Fig. S4). Finally, $PM_{2.5}$ levels are underestimated by up to 10
$\mu gm^{-3}$ over most of Europe (Fig. 4d), with larger underestimations over the eastern Europe up
to 30 $\mu gm^{-3}$.
Over North America, the hourly $O_3$ variation is well captured by all models (Table S1), with
DK1 having slightly lower $r$ coefficient compared to the other models and largest *NMB* (Fig.
3a). The hourly variation of CO and $SO_2$ levels are simulated with relatively lower $r$ values
(Figs. 3b, c), with $SO_2$ levels having the highest underestimations. The $PM_{2.5}$ levels are
underestimated by ~15% except for the DE1 model, having a large underestimation of 63%
(Table S1). As DE1 and US3 use the same SMOKE emissions and CTM, the large difference
in $PM_{2.5}$ concentrations can be partly due to the differences in horizontal and vertical
resolutions in the model setups, as can also be seen in the differences in the CO
concentrations. There are also differences in the aerosol modules and components that each
model simulates. For example, DE1 uses an older version of the secondary organic aerosol
(SOA) module, producing ~3 $\mu gm^{-3}$ less SOA, which can explain ~20% of the bias over
North America. Over the North American domain, the median outscores the mean for $O_3$ ( by
35%), CO (by 52%) and $PM_{2.5}$ (by 29%) while for $SO_2$, the median produces 26% higher
*NMB* compared to the mean. DK1 model simulates a much higher bias for $O_3$ and $SO_2$
compared to other models in the North American domain, while DE1 has the largest bias for
CO and $PM_{2.5}$.
DM8H $O_3$ levels are generally underestimated by the MM mean over the U.S. by up to 20
ppb, while over the eastern and central U.S. there are also overestimations by up to 10 ppb
(Fig. 5a). As seen in Fig. S6, all three models have very similar performance over the U.S.,
with DK1 simulating a slightly lower underestimation and a higher overestimation compared
to DE1 and US3. DE1 and DK1 have very similar spatial pattern in terms of CO bias, in
particular over the eastern coast of the U.S. (Fig. S7). CO levels are underestimated by ~100
ppb over majority of the stations, especially over the eastern U.S., while there are much
larger underestimation over the western U.S. by up to 1000 ppb (Fig. 5b). $SO_2$ levels are
underestimated by up to 5 ppb over the majority of the stations in the U.S., with few
overestimations of up to 5 ppb (Fig. 5c). DE1 and DK1 have very similar spatial distribution
of bias, while US3 has slightly more overestimations (Fig. S8) Finally, PM2.5 levels are
underestimated over majority of the stations by up to 6 $\mu gm^{-3}$, with few overestimations by 2-
4 $\mu gm^{-3}$ (Fig. 5d). DE1 has the largest underestimations compared to DK1 and US3 (Fig. S9).
Table S1 shows that the ensemble median performs slightly better than the ensemble mean
for all pollutants over both continents regarding the bias and error, while the difference on $r$
is rather small. Over the European stations, the median has improved results over the mean
by up to 14% for $r$ and up to 9% for the *RMSE*. The improvements in $r$ over the U.S. are
much smaller compared to Europe (up to ~4%), while the *RMSE* is improved by up to 27%,
except for $SO_2$ where the median has 14% higher *RMSE* than the mean.
3.2. Health outcomes and their economic valuation in Europe
The different health outcomes calculated by each model in Europe as well as their multi
model mean and median are presented in Table S2. Table 3 presents the mean of the
individual model estimates as $MM_{mi}$. Standard deviations calculated from the individual
model estimates are presented along with the $MM_{mi}$ in the text. The health impact estimates
vary significantly between different models. The different estimates obtained are found to
vary up to a factor of three.  Among the different health outcomes, the individual models
simulated the number of congestive heart failure (CHF) cases to be between 19 000 to 41 000
(mean of all individual models, $MM_{mi}$, 31 000 ± 6 500). The number of lung cancer cases due
to air pollution are calculated to be between 30 000 to 78 000 (mean of all individual models,
$MM_{mi}$, 55 000 ± 14 000). Finally, the total (acute + chronic) number of premature death due
to air pollution is calculated to be 230 000 to 570 000 (mean of all individual models, $MM_{mi}$,
414 000 ± 100 000). The health impacts calculated as the median of individual models differ
slightly (~±1%) from those calculated as the mean of individual models (Table S2) due to the
slight differences in the model bias (*NMB*) and error (*NMGE* and *RMSE*) between the mean
and the median performance statistics of the models.
In addition to averaging the health estimates from individual models ($MM_{mi}$), we have also
produced a multi-model mean concentration data ($MM_m$) by taking the average of
concentrations of each species calculated by all models at each grid cell and hour, and fed it
to the EVA model. We have calculated the number of premature death cases in Europe as
410 000 (Table 3) using $MM_m$. Difference between the health impacts calculated using $MM_m$
data from the mean of all individual model ($MM_{mi}$) estimates is smaller than 1%. The number
of premature death cases in Europe as calculated as the average of all models in the multi
model ensemble, $MM_{mi}$, due to exposure to $O_3$ is 12 000 ± 6 500, while the cases due to
exposure to $PM_{2.5}$ is calculated to be 390 000 ± 100 000 [180 000 – 550 000]. The $O_3$-related
mortality well agrees with Liang et al. (2017) that used the multi-model mean of the HTAP2
global model ensemble, which calculated an $O_3$-realted mortality of 12 800 [600 - 28 100].
The multi-model mean ($MM_{mi}$) $PM_{2.5}$-related mortality in the present study is much higher
than that from the HTAP2 study (195 500 [4 400 – 454 800]). The results also agree with the
most recent EEA findings (EEA, 2015), which calculated a total premature death of 419 000
die to $O_3$ and $PM_{2.5}$ in the EU-28 countries. There is also agreement with Geels et al. (2015)
that calculated 388 000 premature death cases in Europe for the year 2000. This difference
can be attributed to the number of mortality cases as calculated by the individual models,
where the HTAP2 ensemble calculates a much lower minimum while the higher ends from
the two ensembles well agree.
The differences between the health outcomes calculated by the HTAP2 and AQMEII
ensembles arise firstly from the differences in the concentrations fields due to the differences
in models, in particular spatial resolutions as well as the gas and aerosols treatments in
different models, but also the differences in calculating the health impacts from these
concentrations fields. EVA calculates the acute premature death due to $O_3$ by using the
SOMO35 metric. On the other hand, in HTAP2 $O_3$-related premature death is calculated by
using the 6-month seasonal average of daily 1-h maximum $O_3$ concentrations. Both groups
use the annual mean $PM_{2.5}$ to calculate the $PM_{2.5}$-related premature death. In addition to $O_3$
and $PM_{2.5}$, EVA also takes into account the health impacts from CO and $SO_2$, which is
missing in the HTAP2 calculations.
Among all models, DE1 model calculated the lowest health impacts for most health
outcomes, which can be attributed to the largest underestimation of $PM_{2.5}$ levels ($NMB$=-
63%: Table S2) due to lower spatial resolution of the model that dilutes the pollution in the
urban areas, where most of the population lives. The number of premature deaths calculated
by this study is in agreement with previous studies for Europe using the EVA system (Brandt
et al., 2013a; Geels et al., 2015). Recently, EEA (2015) estimated that air pollution is
responsible for more than 430 000 premature deaths in Europe, which is in good agreement
with the present study.
Fig. 6a. presents the geographical distribution of the number of premature death in Europe in
2010. The figure shows that the numbers of cases are strongly correlated to the population
density (Fig. 1a), with the largest numbers seen in the Benelux and the Po Valley regions that
are characterized as the pollution hot spots in Europe as well as in megacities such as
London, Paris, Berlin and Athens.
The economic valuation of the air pollution-associated health impacts calculated by the
different models along with their mean and median are presented in Table 4. A total cost of
196 to 451 billion Euros (MM mean cost of $300 \pm 70$ billion Euros) was estimated over
Europe (EU28). Results show that 5% [1% - 11%] of the total costs is due to exposure to $O_3$,
while 89% [80% - 96%] is due to exposure to $PM_{2.5}$. Brandt et al. (2013a) calculated a total
external cost of 678 billion Euros for the year 2011 for Europe, larger than the estimates of
this study, which can be explained by the differences in the simulation year and the emissions
used in the models as well as the countries included in the two studies (the previous study
includes e.g. Russia).
3.3. Health outcomes and their economic valuation in the U.S.
The different health outcomes calculated by each model for the U.S. as well as their mean
and median are presented in Table S2. The variability among the models (~3) is similar to
that in Europe.  The number of congestive heart failure cases in the U.S. as calculated as the
average of all models in the ensemble ($MM_{mi}$) is calculated to be 13 000 [7 000 – 18 000],
while the lung cancer cases due to air pollution are calculated to be 22 000 [9 000 – 31 000].
Finally, the number of premature deaths due to air pollution is calculated to be $165\,000 \pm$
75 000, where $25\,000 \pm 6\,000$ cases are calculated due to exposure to $O_3$ and $140\,000 \pm 72$
000 cases due to exposure to $PM_{2.5}$. The $MM_m$ dataset leads to a number of premature death
of 149 000 that is 6% smaller than the average estimate from individual models ($MM_{mi}$). Due
to the large reduction of $NMB$ by the median compared to the mean of individual models
(Table S1), the multi-model health impacts calculated as the median of health impacts from
individual models are ~13% higher than the health impacts calculated from the $MM_{mi}$. The
$O_3$- and $PM_{2.5}$ mortality cases as calculated by the AQMEII and HTAP2 model ensembles
reasonably agree. Liang et al. (2017) calculated an $O_3$-related mortality of 14 700 [900 –
30 400] and a $PM_{2.5}$-related mortality of 78 600 [4 500 – 162 600]. These results are in very
good agreement with the U.S. EPA (2011) estimates of number of premature death cases of
160 000 in year 2010 and with Caizzo et al. (2013), who calculated 200 000 premature death
cases from combustion sources in the U.S. Among all models, DE1 model calculated the
lowest health impacts for most health outcomes, which can be attributed to the largest
underestimation of $PM_{2.5}$ levels ($NMB$=-63%: Table S2).
The premature death cases in North America are mostly concentrated over the New York
area, as well as in hot spots over Chicago, Detroit, Houston Los Angeles and San Francisco
(Fig. 6b). The figure shows that the number of cases is following the pattern of the population
density. The economic valuation of the air pollution-associated health impacts calculated by
the different models in the U.S. are shown in Table 4. As seen in the table, a total cost of
~145 billion Euros is calculated. Results show that ~22% of the total costs is due to exposure
to $O_3$ while ~78% is due to exposure to $PM_{2.5}$. The major health impacts in terms of their
external costs are slightly different in North America compared to Europe.
3.4. Health impacts and their economic valuation through optimal reduced ensemble subset
The effect of pollution concentrations (EVA input) on health impacts (EVA output) is
investigated in order to estimate the contribution of each air pollutant in the EVA system to
health impacts over different concentration levels. The technical details are provided in the
supplement.
Results show that for the particular input (gridded air pollutant concentrations from
individual model)-output (each health outcome) configuration, the $PM_{2.5}$ drives the variability
of the different health impact and that at least 81% of the variation of the health impacts are
explained by sole variations in the pollutants (i.e. without interactions: Table S3). Table S1
also shows that the most important contribution to the health impacts is from $PM_{2.5}$, followed
by CO and $O_3$ (with much smaller influence though). The impact  of perturbing $PM_{2.5}$ by a
fixed fraction of its standard deviation on the health impact is roughly double compared to
CO and $O_3$.
We have run the EVA system over an all-models mean ($MM_m$) dataset and an optimal
reduced ensemble dataset ($MM_{opt}$) calculated for each of the pollutants in the two domains in
order to see how and whether an optimal reduced ensemble changes the assessment of the
health impacts compared to an all- models ensemble mean. Table 5 shows some sensible
error reduction, although the temporal and spatial averages mask the effective improvement
in accuracy from $MM_m$ to $MM_{opt}$. In Europe, the optimal reduced ensemble decreases the
RMSE by up to 24%, while in NA, the error reduction is much larger (4% to up to 147%). On
a seasonal basis, $MM_{opt}$ reduces $RMSE$ in $PM_{2.5}$ over Europe by 23% in winter while smaller
decreases are achieved in other seasons (~10%). Regarding $O_3$, improvement is 16%-22%,
with the largest improvement in spring. In NA, the improvement in winter $RMSE$ in $PM_{2.5}$ is
smallest (~2%) while larger improvements are achieved in other seasons (~7% - ~9%). For
O$_3$, the largest *RMSE* reduction in NA is achieved for the summer period by 14%.
The analysis of the aggregated health indices data for Europe (Table S1) shows that EVA
indices rely principally on the PM$_{2.5}$ levels and then the CO and O$_3$ values. Therefore, the
relative improvement of the indices with the optimal ensemble should be proportional to the
relative improvement in PM$_{2.5}$, CO and O$_3$. The proportionality rate for each pollutant is
given in Table S3, assuming all pollutants are varied (from *MM$_m$* to *MM$_{opt}$*) away from their
mean by the same fraction of their variance. As seen in the Table 3, from *MM$_m$* to *MM$_{opt}$*, the
health indices increase by up to 30% in Europe. This increase is due to a 27% increase in the
domain mean PM$_{2.5}$ levels when the optimal reduced ensemble is used, as well a slight
increase in O$_3$ by ~1%. The number of premature deaths in Europe increase from 410 000 to
524 000 (28%), resulting in a much higher estimate compared to previous mortality studies.
On the contrary, in the U.S., the mean PM$_{2.5}$ and O$_3$ levels decrease from 2.94 µg m$^{-3}$ to 2.62
µgm$^{-3}$ (~11%) and 18.7 ppb to 18.4 ppb (~2%), respectively. In response, the health indices
decrease by ~11% (Table 3). The number of premature death cases in NA decrease from
149 000 to 133 000.
3.5. Impact of anthropogenic emissions on the health impacts and their economic valuation
The impacts of emission perturbations on the different health outcomes over Europe and the
U.S. as calculated by the individual models are presented in Tables S4-S6. Table 6 shows the
impacts of the different emission perturbations on the premature death cases in Europe and
the U.S as calculated by a subset of models that simulated the base case and all three
perturbation scenarios (*MM$_c$*). Results show that in Europe, the 20% reduction in the global
anthropogenic emissions leads to ~17% domain-mean reduction in all the health outcomes,
with a geographical variability as seen in Fig. 6c. The figure shows that the larger changes in
mortality is calculated in the central and northern parts of Europe (15-20% decreases), while
the changes are smaller in the Mediterranean region (5-10%), highlighting the non-linearity
of the response to emission reductions. However, it should be noted that global models or
coarse-resolution regional models (as in this study) cannot capture the urban features and
pollution levels and thus, non-linearities should be addressed further using fine spatial
resolutions or urban models. The models vary slightly simulating the response to the 20%
reduction in global emissions, estimating decreases of ~11% to 20%. The number of
premature deaths decreased on average by ~50 000, ranging from -39 000 (DK1) to -103 000
(IT1). This number is in good agreement with the ~45 000 premature death calculated by the
HTAP2 global models (Liang et al., 2017). The *MM$_c$* ensemble calculated a 15% and 17%
decrease in the O$_3$- and PM$_{2.5}$-related premature death cases, respectively, in response to the
GLO scenario. This decrease in the global anthropogenic emissions leads to an estimated
decrease of 56 ± 18 billion Euros in associated costs in Europe (Table 6).
As seen in Table 8, a 20% reduction of anthropogenic emissions in the EUR region, as
defined in HTAP2, avoids 47 000 premature death, while a 20% reduction of the
anthropogenic emissions in the NAM region leads to a much smaller decrease of premature
deaths in Europe (~1 000). These improvements in the number of premature deaths are in
agreement with a recent HTAP2 global study that calculated reductions of ~34 000 and
~1 000 for the EUR and NAM scenarios, respectively (Liang et al., 2017) and with Anenberg
et al. (2009 and 2014), which totals to a sum of avoided premature deaths being ~39 000 and
1 800 as calculated by the MM mean. Both the global and regional models agree that the
largest impacts of reducing emissions with respect to premature deaths come from emission
within the source region, while foreign sources contribute much less to improvements in
avoiding adverse impacts of air pollution. The decreases in health impacts in EUR and NAM
scenarios corresponds to decreases in the associated costs by -47 ± 16 billion Euros and -1.4
± 0.4 billion Euros, respectively. This is consistent with results in Brandt et al. (2012), where
a contribution of ~1% to $PM_{2.5}$ concentrations in Europe is originating from the NAM region.
The 20% reduction in global anthropogenic emissions leads to 18% reduction in the health
outcomes (Table 8) in the U.S., with a geographical variability in the response. Fig. 6d shows
that the largest decreases in mortality is calculated for the western coast of the U.S. (~20%)
and slightly lower response in the central and eastern parts of the U.S. (15-20%). The number
of premature death cases, as calculated by the mean of all individual models decreases from
~160 000 ± 70 000 to ~130 000 ± 60 000, avoiding 24 ± 10 billion Euros (Table 6) in
external costs, also in agreement with the ensemble of HTAP2 global models (~23 000) The
$O_3$-related premature death cases decreased by 42% while the $PM_{2.5}$-relared cases decreased
by 18%.
A 20% reduction of the North American emissions avoids ~25 000 ± 12 000 premature
deaths (-16%), suggesting that ~80% of avoided premature deaths are achieved by reductions
within the source region while 20% (~5 000 premature deaths) is from foreign sources. This
number is also in good agreement with Liang et al. (2017) that estimated a reduction of
premature deaths of ~20 000 due to $O_3$ and $PM_{2.5}$ in the United States due to an emission
reduction of 20% within the region itself, using the ensemble mean of the HTAP2 global
models. These results are much larger than the number of avoided premature death of
~11 000 as calculated by the sum of Anenberg et al. (2009 and 2104).The corresponding
benefit is calculated to be 21 ± 9 billion Euros in the NAM scenario. According to results
from the EAS scenario, among these 5 000 avoided cases that are attributed to the foreign
emission sources, 1 900 ± 2 000 premature deaths can be avoided by a 20% reduction of the
East Asian emissions, avoiding 2.5 ± 3 billion Euros. Our number of avoided premature
deaths due to the EAS scenario is much higher than 580 avoided premature deaths calculated
by Liang et al. (2017) and 380 avoided cases as calculated by Anenberg et al. (2009 and
733 2014).

**Conclusions**
The impact of air pollution on human health and their economic valuation for the society
across Europe and the United States is modelled by a multi-model ensemble of regional
models from the AQMEII3 project. All regional models used boundary conditions from the
C-IFS model, and emissions from either the MACC inventory in Europe or the EPA
inventory for the North America, or the global inventory from HTAP. Sensitivity analysis on
the dependence of models on different sets of boundary conditions has not been conducted so

far but large deviations from the current results in terms of health impacts are not expected. The modelled surface concentrations by each individual model are used as input to the EVA system to calculate the resulting health impacts and the associated external costs from $O_3$, $CO$, $SO_2$ and $PM_{2.5}$. Along with a base case simulation for the year 2010, some groups performed additional simulations, introducing 20% emission reductions both globally and regionally in Europe, North America and East Asia.

The base case simulation of each model is evaluated with available surface observations in Europe and North America. Results show large variability among models, especially for $PM_{2.5}$, where models underestimate by ~20% - ~60%, introducing a large uncertainty in the health impact estimates as $PM_{2.5}$ is the main driver for health impacts. The differences in the models are largely due to differences in the spatial and vertical resolutions, meteorological inputs, inclusion of natural emissions, dust in particular, as well as missing or underestimated SOA mass, which is critical for the $PM_{2.5}$ mass. As shown in the supplementary material, the CTMs diverge a lot on the representation of particles and their size distribution, SOA formation, as well as the inclusion of natural sources. As the anthropogenic emissions are harmonized in the models, they represent a minor uncertainty in terms of model-to-model variation. However, differences in the treatment of the temporal, vertical and chemical distributions of the particulate and volatile organic species have an influence in the model calculations and therefore lead to model-to-model variations.

The variability of health impacts among the models can be up to a factor of three in Europe (twelve models) and the U.S. (three models), among the different health impacts. The multi-model mean total number of premature death is calculated to be 414 000 in Europe and 160 000 in the U.S., where $PM_{2.5}$ contributes by more than 90%. These numbers agree well with previous global and regional studies for premature deaths due to air pollution. In order to reduce the uncertainty coming from each model, an optimal ensemble set is produced, that is, the subset of models that produce the smallest error compared to the surface observations at each time step. The optimum ensemble results in an increase of health impacts by up to 30% in Europe and a decrease by ~11% in the United States. These differences clearly demonstrate the importance of the use of optimal-reduced multi-model ensembles over traditional all model-mean ensembles, both in terms of scientific results, but also in policy applications.

Finally, the role of domestic versus foreign emission sources on the related health impacts is investigated using the emission perturbation scenarios. A global reduction of anthropogenic emissions by 20% decreases the health impacts by 17%, while the reduction of foreign emissions decreases the health impacts by less than 1%. The decrease of emissions within the source region decreases the health impacts by 16%. These results show that the largest impacts of reducing emissions with respect to the premature death come from emissions within the source region, while foreign sources contributing to much less improvements in avoiding adverse impacts of air pollution.

**Outlook**

Currently health assessments of airborne particles are carried out under the assumption that all fine fraction particles affect health to a similar degree independent of origin, age and chemical composition of the particles. A 2013 report from WHO concludes that the cardiovascular effects of ambient $PM_{2.5}$ are greatly influenced, if not dominated, by their transition metal contents (WHO, 2013b). It is known that trace metals and traffic markers are highly associated with daily mortality (Lippmann, 2014). Even low concentrations of trace metals can be influential on health related responses.

Regarding ambient concentrations of PM and the exposure-response functions (ERFs), there is a rich set of studies providing information on total PM mass. However, only few studies focus on individual particulate species, mainly black carbon and carbonaceous particles. In addition to PM, studies on human populations have not been able to isolate potential effects of $NO_2$, because of its complex link to PM and $O_3$. The WHO REVIHAAP review from 2013 concludes that health assessments based on $PM_{2.5}$ ERFs will be most inclusive (WHO, 2013b). In addition, the ERFs are based on urban background measurements, introducing uncertainties regarding non-urban areas or high pollution areas as e.g. street canyons. Current state-of-the-art health impact estimates, in particular on regional to global scales, assume a correlation with exposure to outdoor air pollution, while in reality, exposure is dynamic and depends on the behavior of the individual. In addition, differences in age groups, gender, ethnicity and behavior should be considered in the future studies. There are also uncertainties originating from the representations of the aerosols in the atmospheric models used in the calculation of pollutant concentrations as well as the emissions. Further developments in the aerosol modules, such as the representation of organic aerosols and windblown and suspended dust, are need in order to achieve mass closure of PM to get robust estimates of health impacts. In addition, new findings show that $O_3$ has also chronic health impacts in addition to its acute impacts (WHO, 2013a; Turner et al., 2016).

Due to above reasons, there is a large knowledge gap regarding the health impacts of particles. There are a number of ongoing projects trying to identify the health impacts from individual particle components and produce individual ERFs for these components. NordicWelfAir project (http://projects.au.dk/nordicwelfair/) aims to investigate the potential causal impact of individual chemical air pollutants as well as mixtures of air pollutants on health outcomes. In pursuing this aim, the project uses the unique Nordic population-based registers allowing linkage between historical residential address, air pollutants over decades and later health outcomes. By linking the exposure to health outcomes, new exposure-response relationships can be determined of health effects for different population groups (e.g. age, education, ethnicity, gender, lifestyle, and working life vs. retirement conditions) related to air pollution for the individual chemical air pollutants. In addition, the high resolution simulations conducted will enable us to have a better understanding of non-linearities between the emissions, health impacts, and their economic valuation.

## ACKNOWLEDGEMENTS

We gratefully acknowledge the contribution of various groups to the third air Quality Model Evaluation international Initiative (AQMEII) activity. Joint Research Center Ispra/Institute

for Environment and Sustainability provided its ENSEMBLE system for model output
harmonization and analyses and evaluation. Although this work has been reviewed and
approved for publication by the US Environmental Protection Agency, it does not necessarily
reflect the views and policies of the agency. Aarhus University gratefully acknowledges the
NordicWelfAir project funded by the NordForsk's Nordic Programme on Health and Welfare
(grant agreement no. 75007), the REEEM project funded by the H2020-LCE Research and
Innovation Action (grant agreement no.: 691739), and the Danish Centre for Environment
and Energy (AU-DCE). University of L'Aquila thanks the EuroMediterranean Center for
Climate Research (CMCC) for providing the computational resources. RSE contribution to
this work has been financed by the research fund for the Italian Electrical System under the
contract agreement between RSE S.p.A. and the Ministry of Economic Development –
General Directorate for Nuclear Energy, Renewable Energy and Energy Efficiency in
compliance with the decree of 8 March 2006.

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

Table 1.Key features (meteorological/chemistry and transport models, emissions, horizontal and vertical grids) of the regional models participating to the AQMEII3 health impact study and the perturbation scenarios they performed.

| Group Code | Model | Emissions | Horizontal Resolution | Vertical Resolution | Gas Phase | Aerosol Model | Europe | | | | North America | | | |
|---|---|---|---|---|---|---|---|---|---|---|---|---|---|---|
| | | | | | | | BASE | GLO | NAM | EUR | BASE | GLO | EAS | NAM |
| DE1 | COSMO-CLM/CMAQ | HTAP | 24 km × 24 km | 30 layers, 50 hPa | CB5-TUCL | 3 modes | × | × | × | × | × | × | × | × |
| DK1 | WRF/DEHM | HTAP | 50 km × 50 km | 29 layers, 100 hPa | Brandt et al. (2012) | 2 modes | × | × | × | × | × | × | × | × |
| ES1 | WRF/CHEM | MACC | 23 km × 23 km | 33 layers, 50 hPa | RADM2 | 3 modes, MADE/SORGAM | × | | × | | | | | |
| FI1 | ECMWF/SILAM | MACC | 0.25° × 0.25° | 12 layers, 13 km | CB4 | 1-5 bins, VBS | × | × | × | × | | | | |
| FRES1 | ECMWF/CHIMERE | HTAP | 0.25° × 0.25° | 9 layers, 50 hPa | MELCHIOR2 | 8 bins | × | × | × | × | | | | |
| IT1 | WRF/CHEM | MACC | 23 km × 23 km | 33 layers, 50 hPa | RACM-ESRL | 3 modes, MADE/VBS | × | × | | × | | | | |
| IT2 | WRF/CAMx | MACC | 23 km × 23 km | 14 layers, 8 km | CB5 | 3 modes | × | × | | | | | | |
| NL1 | LOTOS/EUROS | MACC | 0.50° × 0.25° | 4 layers, 3.5 km | CB4 | 2 modes, VBS | × | | | | | | | |
| TR1 | WRF/CMAQ | MACC | 30 km × 30 km | 24 layers, 10hPa | CB5 | 3 modes | × | × | × | | | | | |
| UK1 | WRF/CMAQ | MACC | 15 km × 15 km | 23 layers, 100 hPa | CB5-TUCL | 3 modes | × | × | × | × | | | | |
| UK2 | WRF/CMAQ | HTAP | 30 km × 30 km | 23 layers, 100 hPa | CB5-TUCL | 3 modes | × | × | | | | | | |
| UK3 | WRF/CMAQ | MACC | 18 km × 18 km | 35 layers, 16 km | CB5 | 3 modes | × | × | × | | | | | |
| US3 | WRF/CMAQ | SMOKE | 12 km × 12 km | 35 layers, 50 hPa | CB5-TUCL | 3 modes | | | | | × | × | × | × |

Table 2. Exposure-response functions, the concentrations metrics, and economic valuations used in the EVA model.

| Health effects (compounds) | Exposure-response coefficient | Valuation, €[2013] |
|---|---|---|
| | (α) | (EU27 & NA) |
| **Morbidity** | | |
| Chronic Bronchitis[1], CB (PM) | 8.2E-5 cases/$\mu gm^{-3}$ (adults) | 38,578 per case |
| Restricted activity days[2], RAD (PM) | =8.4E-4 days/ $\mu gm^{-3}$ (adults) | 98 per day |
| | -3.46E-5 days/ $\mu gm^{-3}$ (adults) | |
| | -2.47E-4 days/ $\mu gm^{-3}$ (adults>65) | |
| | -8.42E-5 days/ $\mu gm^{-3}$ (adults) | |
| Congestive heart failure[3], CHF (PM) | 3.09E-5 cases/ $\mu gm^{-3}$ | 10,998 per case |
| Congestive heart failure[3], CHF (CO) | 5.64E-7 cases/ $\mu gm^{-3}$ | |
| Lung cancer[4], LC (PM) | 1.26E-5 cases/ $\mu gm^{-3}$ | 16,022 per case |
| **Hospital admissions** | | |
| Respiratory[5], RHA (PM) | 3.46E-6 cases/ $\mu gm^{-3}$ | 5,315 per case |
| Respiratory[5], RHA (SO2) | 2.04E-6 cases/ $\mu gm^{-3}$ | |
| Cerebrovascular[6], CHA (PM) | 8.42E-6 cases/ $\mu gm^{-3}$ | 6,734 per case |
| **Asthma children (7.6 % < 16 years)** | | |
| Bronchodilator use[7], BUC (PM) | 1.29E-1 cases/ $\mu gm^{-3}$ | 16 per case |
| Cough[8] – COUC (PM) | 4.46E-1 days/ $\mu gm^{-3}$ | 30 per day |
| Lower respiratory symptoms[7], LRSA (PM) | 1.72E-1 days/ $\mu gm^{-3}$ | 9 per day |
| **Asthma adults (5.9 % > 15 years)** | | |
| Bronchodilator use[9], BUA (PM) | 2.72E-1 cases/ $\mu gm^{-3}$ | 16 per case |
| Cough[9], COUA (PM) | 2.8E-1 days/ $\mu gm^{-3}$ | 30 per day |
| Lower respiratory symptoms[9], LRSA (PM) | 1.01E-1 days/ $\mu gm^{-3}$ | 9 per day |
| **Mortality** | | |
| Acute mortality[10,11] (SO2) | 7.85E-6 cases/ $\mu gm^{-3}$ | 1,532,099 per case |
| Acute mortality[10,11] (O3) | 3.27E-6*SOMO35 cases/ $\mu gm^{-3}$ | |
| Chronic mortality[4,12], YOLL (PM) | 1.138E-3 YOLL/ $\mu gm^{-3}$ (>30 years) | 57,510 per YOLL |
| Infant mortality[13], IM (PM) | 6.68E-6 cases/ $\mu gm^{-3}$ (> 9 months) | 2,298,148 per case |

[1] Abbey et al. (1995), [2] Ostro (1987), [3] Schwartz and Morris (1995), [4] Pope et al. (2002), [5] Dab et al. (1996), [6] Wordley et al. (1997), [7] Roemer et al. (1993), [8] Pope and Dockerey (1992), [9] Dusseldorp et al. (1995), [10] Anderson (1996), [11] Touloumi (1996), [12] Pope et al. (1995), [13] Woodruff et al. (1997).

Table 3. Health impacts calculated by the mean of individual model estimates (denoted as $MM_{mi}$) and the standard deviation, multi-model mean ensemble without error reduction ($MM_m$) and the optimal ensemble ($MM_{Opt}$) in Europe and the U.S. See Table 2 for the definitions of health impacts. PD stands for premature death. All health impacts are in units of number of cases $\times$ 1000, except for Infant Mortality (IM), which reports directly the number of cases.

| | EU | | | NA | | |
|---|---|---|---|---|---|---|
| | $MM_{mi}$ | $MM_m$ | $MM_{Opt}$ | $MM_{mi}$ | $MM_m$ | $MM_{Opt}$ |
| CB | 360±89 | 360 | 468 | 142±74 | 142 | 125 |
| RAD | 368 266±90 670 | 368245 | 478073 | 145 337±75 250 | 145337 | 127921 |
| RHA | 23±5 | 23 | 28 | 10±4 | 8 | 7 |
| CHA | 46±11 | 46 | 60 | 19±10 | 19 | 16 |
| CHF | 31±6 | 31 | 38 | 13±6 | 9 | 8 |
| LC | 55±14 | 55 | 72 | 22±11 | 22 | 19 |
| BDUC | 10 766±2 650 | 10766 | 13976 | 4 566±2 383 | 4566 | 4019 |
| BDUA | 70 492±17 400 | 70489 | 91511 | 27 819±14 400 | 27819 | 24485 |
| COUC | 37 198±9 160 | 37196 | 48289 | 15 776±8 230 | 15776 | 13886 |
| COUA | 72 566±17 900 | 72562 | 94203 | 28 637±14 830 | 28637 | 25206 |
| LRSC | 14 355±3 530 | 14354 | 18635 | 6 088±3 180 | 6088 | 5359 |
| LRSA | 26 175±6 400 | 26174 | 33980 | 10 330±5 350 | 10330 | 9092 |
| AYOLL | 26±13 | 23 | 20 | 25±7 | 9 | 9 |
| YOLL | 4 111±1 010 | 4111 | 5337 | 1 481±762 | 1481 | 1304 |
| PD | 414±98 | 410 | 524 | 165±76 | 149 | 133 |
| IM* | 403±99 | 403 | 524 | 143±75 | 143.3667 | 126.1 |

Table 4. External costs (in million Euros) related to the health impacts of air pollution as calculated by the individual models over Europe and the United States.

| Models | CO | $SO_2$ | $O_3$ | $PM_{2.5}$ | TOTAL |
|---|---|---|---|---|---|
| Europe | | | | | |
| DE1 | 70 | 19 000 | 22 000 | 155 000 | 196 000 |
| DK1 | 80 | 13 000 | 24 000 | 237 000 | 274 000 |
| ES1 | 70 | 8 000 | 6 000 | 339 000 | 353 000 |
| FI1 | 90 | 18 000 | 5 000 | 335 000 | 358 000 |
| FRES1 | 90 | 15 000 | 13 000 | 305 000 | 333 000 |
| IT1 | 80 | 17 000 | 21 000 | 413 000 | 451 000 |
| IT2 | 70 | 11 000 | 6 000 | 253 000 | 270 000 |
| NL1 | 70 | 12 000 | 18 000 | 215 000 | 245 000 |
| TR1 | 110 | 30 000 | 35 000 | 376 000 | 441 000 |
| UK1 | 80 | 28 000 | 25 000 | 280 000 | 333 000 |
| UK2 | 80 | 34 000 | 27 000 | 340 000 | 401 000 |
| UK3 | 80 | 47 000 | 25 000 | 279 000 | 351 000 |
| MEAN | 81 | 21 000 | 19 000 | 294 000 | 334 000 |
| MEDIAN | 80 | 17 500 | 21 500 | 292 500 | 342 000 |
| The United States | | | | | |
| DE1 | 30 | 9 000 | 21 000 | 46 000 | 76 000 |
| DK1 | 55 | 11 000 | 39 000 | 123 000 | 172 000 |
| US3 | 60 | 14 000 | 22 000 | 155 000 | 191 000 |
| MEAN | 50 | 11 500 | 27 000 | 108 000 | 146 000 |
| MEDIAN | 55 | 11 000 | 22 000 | 123 000 | 172 000 |

Table 5. Annual average RMSE of the multi-model ensemble mean ($MM_m$) and of the optimal reduced ensemble mean ($MM_{opt}$) for the heath impact-related species. Units are ppb for the gaseous species and $\mu g\ m^{-3}$ for $PM_{2.5}$.

| | $O_3$ | | CO | | $SO_2$ | | $PM_{2.5}$ | |
|---|---|---|---|---|---|---|---|---|
| | $MM_m$ | $MM_{opt}$ | $MM_m$ | $MM_{opt}$ | $MM_m$ | $MM_{opt}$ | $MM_m$ | $MM_{opt}$ |
| Europe | | | | | | | | |
| Winter | 10.3 | 8.6 | 502.4 | 490.3 | 6.3 | 5.6 | 22.5 | 20.7 |
| Spring | 12.4 | 9.6 | 247.1 | 239.5 | 4.6 | 3.1 | 9.9 | 7.8 |
| Summer | 13.4 | 10.7 | 197.4 | 188.0 | 3.9 | 2.3 | 8.2 | 5.7 |
| Autumn | 10.7 | 8.8 | 314.5 | 305.5 | 4.6 | 3.1 | 11.0 | 8.7 |
| Annual | 11.7 | 9.4 | 315.3 | 305.8 | 4.8 | 3.5 | 12.9 | 10.7 |
| North America | | | | | | | | |
| Winter | 10.9 | 10.4 | 356.7 | 328.1 | 5.7 | 5.5 | 8.3 | 8.1 |
| Spring | 12.0 | 11.4 | 288.7 | 270.2 | 5.4 | 5.1 | 7.2 | 6.6 |
| Summer | 15.1 | 13.0 | 258.3 | 238.7 | 5.4 | 5.0 | 9.7 | 8.8 |
| Autumn | 12.8 | 11.6 | 330.6 | 307.6 | 5.8 | 5.3 | 7.8 | 7.2 |
| Annual | 12.7 | 11.6 | 308.6 | 286.1 | 5.6 | 5.2 | 8.2 | 7.7 |

Table 6. Impact of the emission reduction scenarios on avoided premature death (ΔPD) and corresponding change in external cost as calculated by the multi-model mean over Europe and the United States.

| Source | Receptor | | | |
|--------|----------|---|---|---|
| | Europe | | The United States | |
| | ΔPD | ΔTotal Cost (billion €) | ΔPD | ΔTotal Cost (billion €) |
| GLO | -54 000 ± 18 000 | -56 ± 18 | -27 500 ± 14 000 | -24 ± 10 |
| NAM | -940 ± 1100 | -1.4 ± 0.4 | -25 000 ± 12 000 | -21 ± 9 |
| EUR | -47 000 ± 24 000 | -47 ± 16 | - | - |
| EAS | - | - | -1 900 ± 2 200 | -2.5 ± 3 |

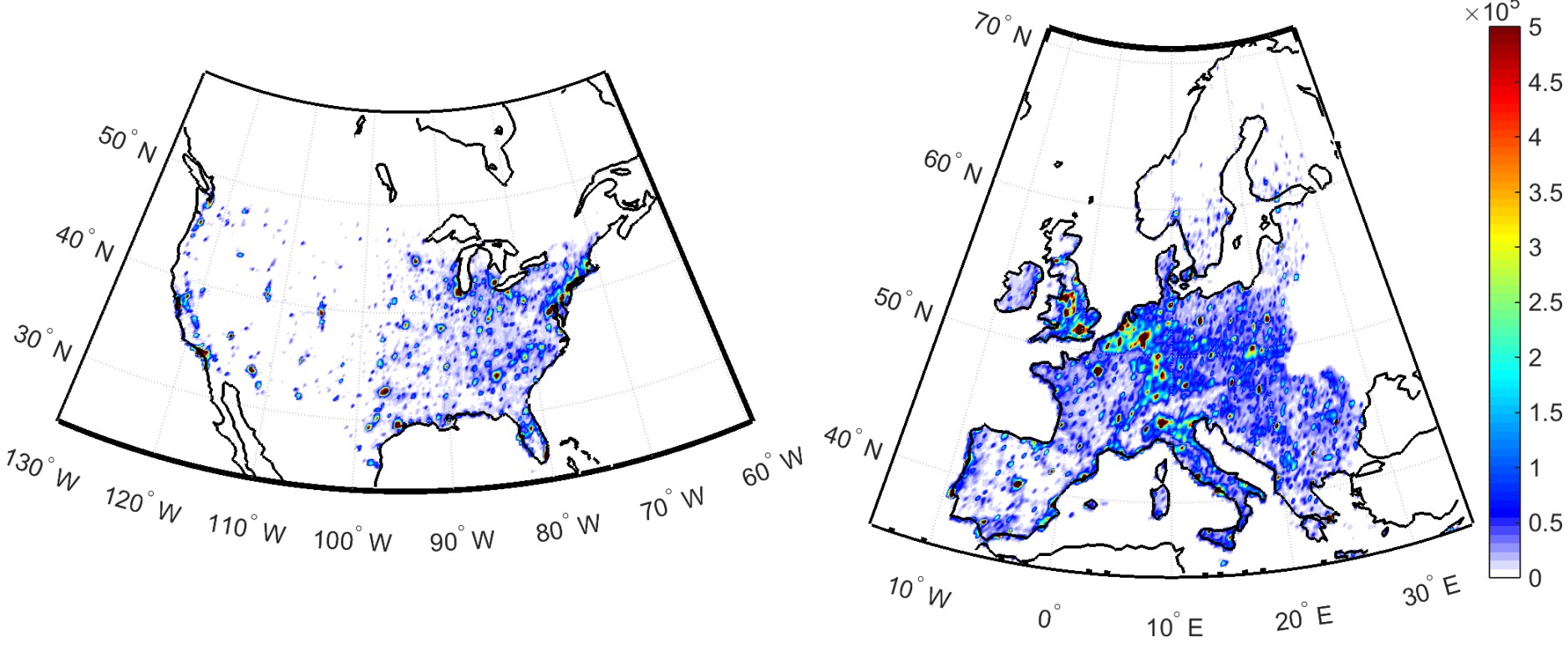

Fig.1. Population density (population per 0.25°×0.25° grid box) over a) the United States and b) Europe.

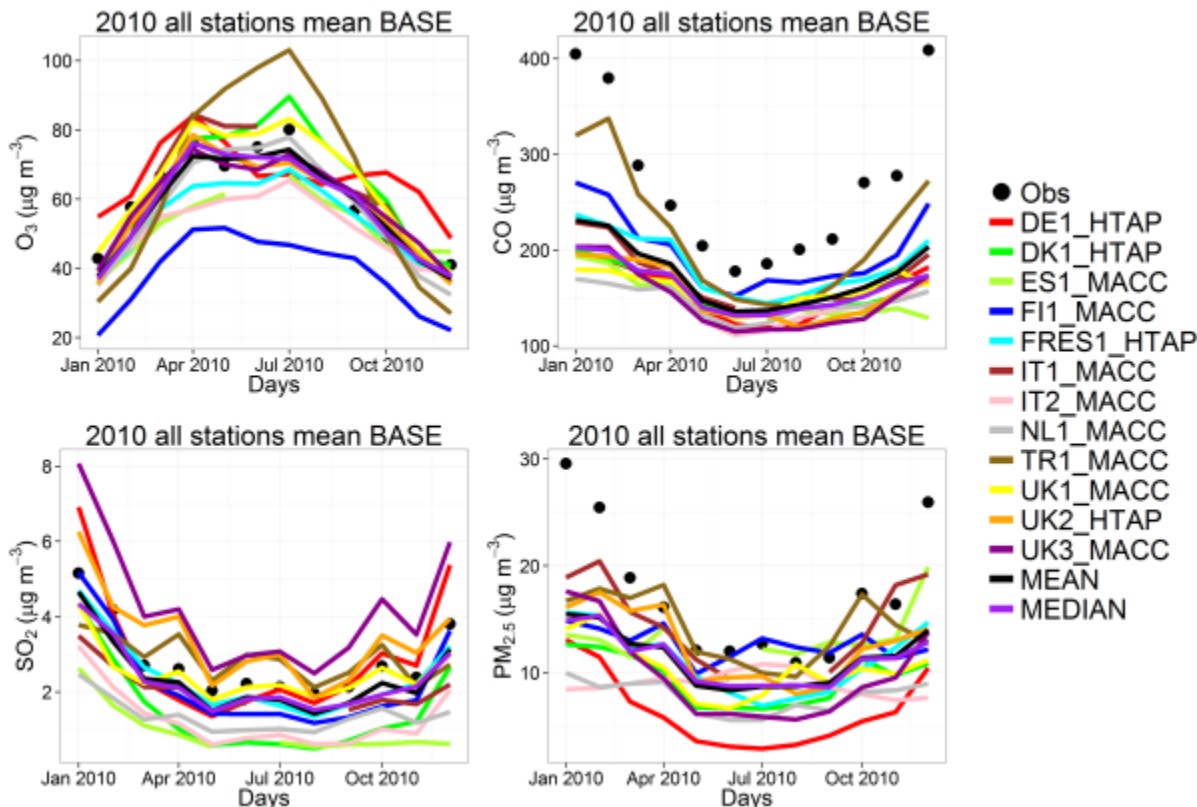

Fig. 2. Observed and simulated (base case) monthly a) $O_3$, b) CO, c) $SO_2$ and d) $PM_{2.5}$ concentrations over Europe.

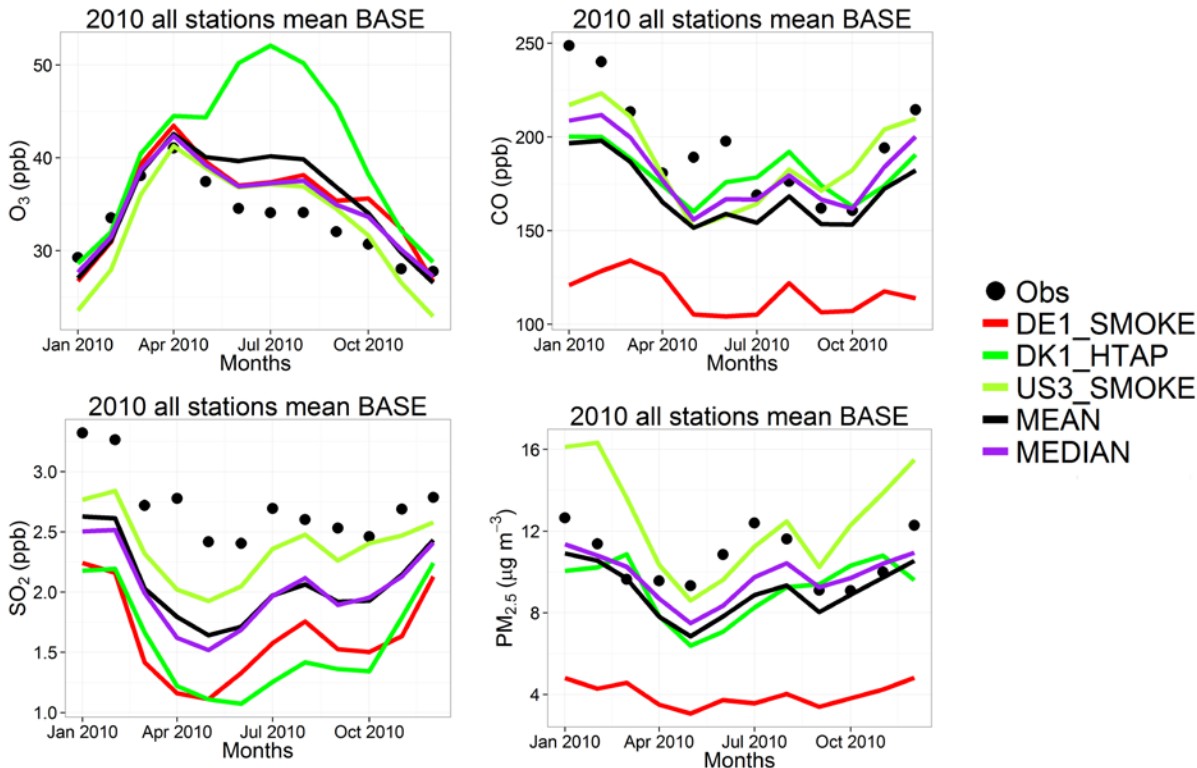

Fig. 3. Observed and simulated (base case) monthly a) $O_3$, b) CO, c) $SO_2$ and d) $PM_{2.5}$ concentrations over the U.S.

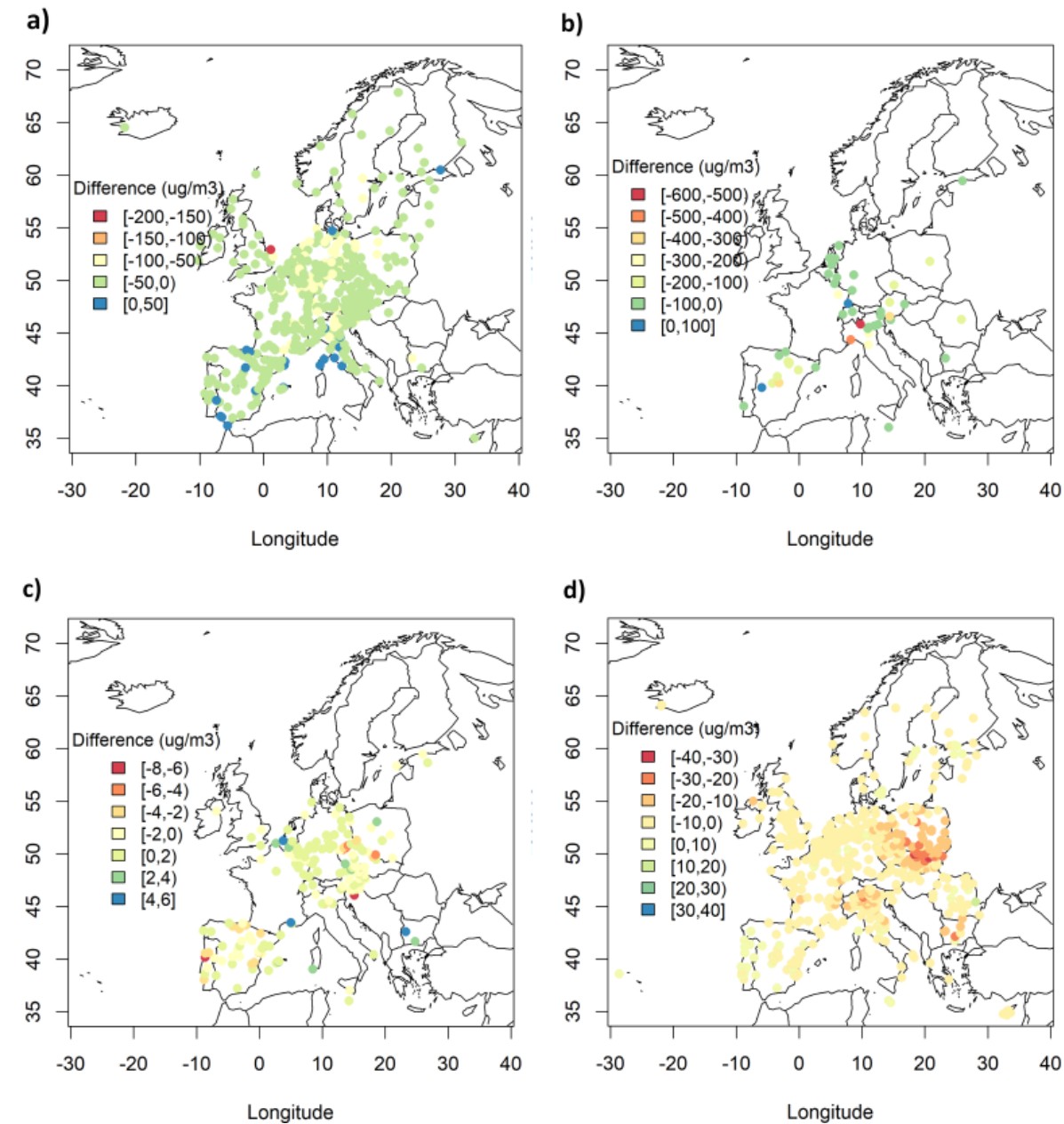

Fig. 4. Spatial distribution of annual MM mean bias ($\mu g m^{-3}$) for a) DM8H $O_3$, b) CO, c) $SO_2$ and d) $PM_{2.5}$ over Europe.

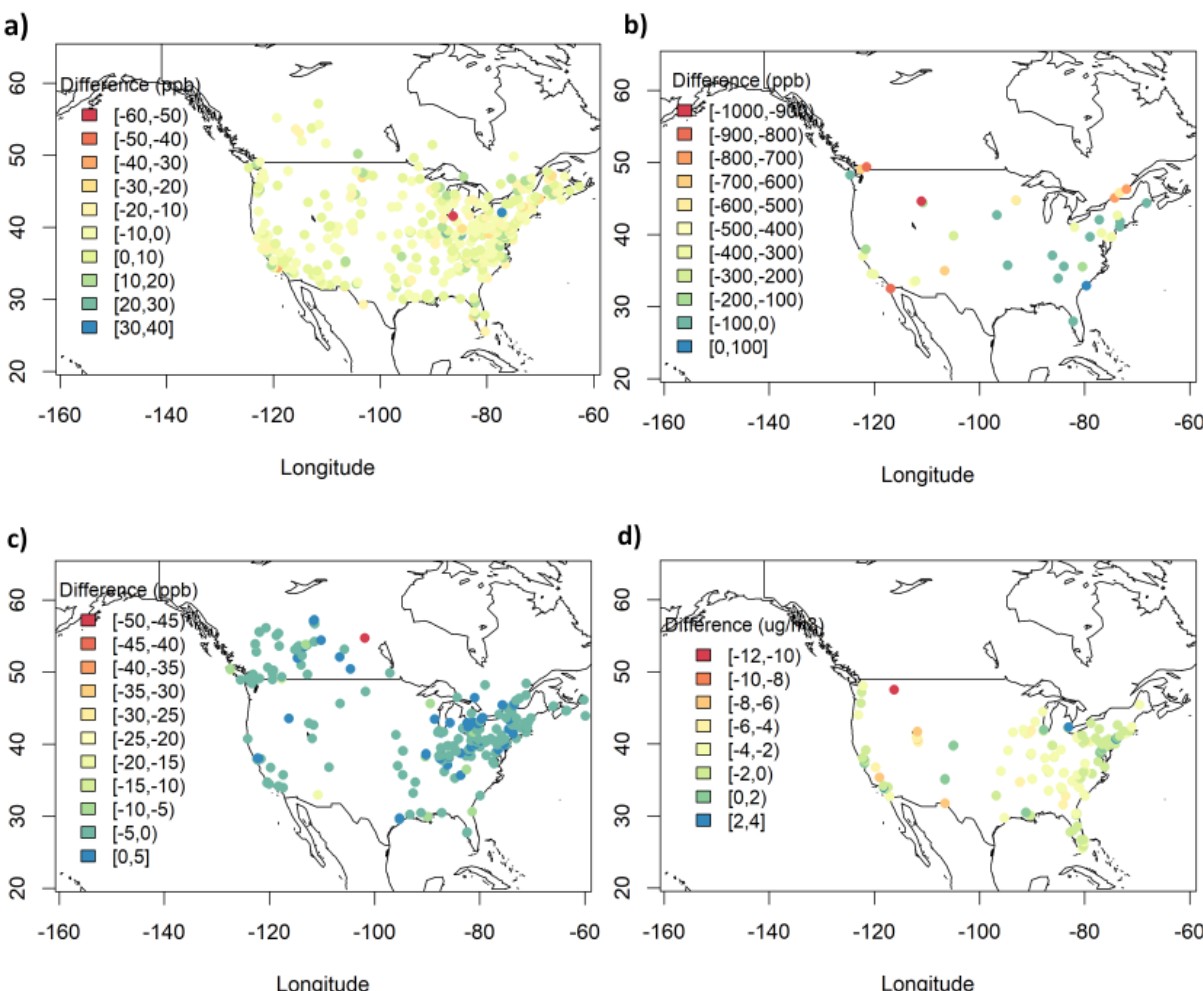

Fig. 5. Spatial distribution of annual MM mean bias (ppb for gases and μgm$^{-3}$ for PM$_{2.5}$) for a) DM8H O$_3$, b) CO, c) SO$_2$ and d) PM$_{2.5}$ over North America.

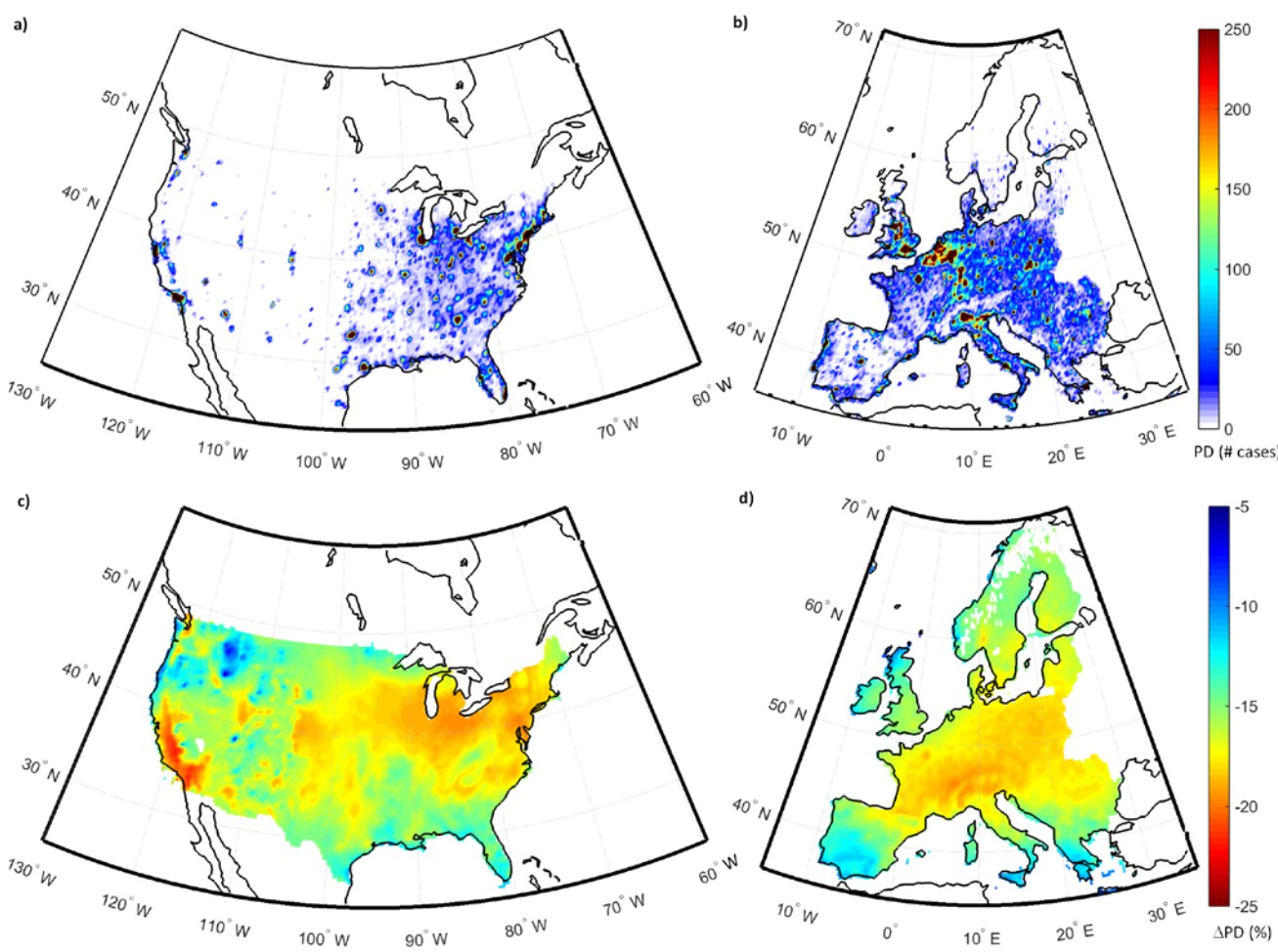

Fig. 6. Spatial distribution of the number of total premature death (PD: units in number of cases per 0.25°×0.25° grid box) in a) the United States and b ) Europe and the relative change (%) in the number of premature death (PD) in response to the GLO scenario in c) the United States and d) Europe in 2010 as calculated by the multi-model mean ensemble.