# Peer review of "Assessment and economic valuation of air pollution impacts on human health over Europe"

_Atmospheric Chemistry and Physics, 2017_

## Referee Comment (RC1) · Anonymous Referee #1 · 22 Nov 2017

This paper uses an ensemble of multiple chemical transport models and health impact functions to estimate the health impacts of air pollution in US and Europe and the impacts of long-range transport of pollution between the two regions. The idea for the paper is interesting and if conducted well it would be a nice contribution to the literature. Unfortunately the implementation is lacking, particularly for the health impact assessment methods and data sources. The authors have generated health impact numbers, it is not clear how they were generated, whether they are supported by the epidemiological evidence, and how they should be interpreted. The analysis of the

model ensemble and different averaging techniques is much stronger. Below are some suggestions mainly for how the health methodology might be improved.

Lines 84-96 should be updated with the most recent GBD 2016 numbers

Lines 118-153 could use some organization. This section is basically just listing results from individual studies without synthesizing them or connecting them to the present study. It's not clear as written by this section is there.

Line 188-190 states that this is the first study to use a common approach for health impact assessment across US and Europe, but the HTAP ozone and PM2.5 health impact assessments referenced earlier used a similar approach. Perhaps the authors are referring only to the economic valuation portion? If so, I'm still not sure this is the first study to do that since there are now several (perhaps many) global health impact and valuation studies that use a common approach for all countries/regions, including US and Europe.

Lines 296-298: given that this paper's focus is on the health impacts, and not the modeling, there should be much more detail given here about the health impact methods in addition to, or instead of, the modeling detail, which can be found in other places and referenced. The health methods quickly summarized here diverge from the methods used by the Global Burden of Disease, U.S. EPA, and many recently published papers. So this needs to be explained, expanded, and justified quite a bit more. As stated, summing ozone deaths with PM2.5 YOLL doesn't make logical sense, as one is cases and one is years, and what is being divided by 10.6 and why? The CAFÉ reference is 12 years old, and air pollution epidemiology and health impact assessment has advanced quite a bit since then. For ozone, there are now studies showing effects of long-term exposure on mortality, just like for PM, so why are only short-term ozone impacts calculated?

Lines 299-302: The ERFs listed in Table 2 are quite a bit out of date, particularly for the U.S. studies. Most of these are 20 years old. There have been many studies now

reporting updated ozone and PM2.5 risk estimates for the American Cancer Society cohort which can be used. And these are not necessarily consistent magnitudes compared with the old studies.

Table 2 needs concentration metrics to which each ERF applies. Section 2.2 should state which concentration metrics were drawn from the models (annual average, annual average of 8-hr daily max, etc.) used which each ERF. I see now these are indicated starting in line 376, but not explained, and should be in section 2.2.

Section 2.2 should also give some equations used to calculate health impacts. It's difficult to understand what was done and impossible to judge whether it's technically sound.

Section 2.2 were the exposure response functions applied in a linear equation or some other functional form (e.g. log-linear)? This is important for the perturbation simulations because you are reducing pollution at the high end, where the shape of the curve can have a big impact on the magnitude of health benefits estimated.

Section 2.2 should also indicate the source of baseline disease rates to calculate health impacts.

Section 2.2 did you first estimate health impacts from each individual model and then average, or first average the concentrations across models and then estimate health impacts?

Section 2.2 what spatial resolution was used to estimate health impacts? Part of the problem with previous studies of PM long-range transport is that the grid resolution was too coarse to adequately capture health benefits from reducing local PM. Spatial scale is important.

Section 3.2 are the plus/minus numbers given with all the results the range of health impacts calculated with individual models? How was uncertainty in the exposure-response function accounted for?

Line 413 appears to be missing a 0 in the HTAP2 result

Line 421 what is meant by "by construction"?

There are many references to the Liang (in preparation) study, but since this study is not yet available the usefulness of these comparisons is limited. It is often used as justification that the present study was done right, since the numbers match up. But there is not currently enough information from either study to judge that.

There are many tables with numbers for health impacts that are difficult to digest. Suggest replacing some of these with figures to highlight the most salient points.

―――――――――――――

---

## Referee Comment (RC2) · Anonymous Referee #2 · 1 Dec 2017

The impact of air pollution on human health and the associated external costs is an interesting topic. The study utilized a multi-model ensemble of regional models to calculate the impacts. The results encourage the use of optimal-reduced multi-model ensembles compared to traditional all model-mean ensembles. The topic is certainly suitable for ACP, the method is appropriate, and the analysis and the results are generally reasonable. This paper can be considered for publication after addressing the following issues.

General comments:

The Abstract is a bit too long. I encourage the authors to shorten their abstract to make it concise and informative. In addition, the authors should be more careful about the units. Many units in the tables and figures are missing or unclear and should be added.

Although the description of the methods is comprehensive, additional description is needed. As the ensemble-contributing members as well as the gridded population density data have different spatial resolutions (see Table 1), the combining methods for those data should be added. Also, what is the spatial resolution of the multi-model ensemble mean ($MM_m$) and the optimal reduced ensemble mean ($MM_{opt}$) (Fig. 4)?

Specific comments:

Line 72: "North American emissions foreign emissions" - delete "foreign emissions".

Line 224-225: "a number of emission perturbation scenarios have been simulated (Table 1)" – there is no EAS emission perturbation scenario for the European domain, and no EUR emission perturbation scenario for the North American domain. Please explain the design of the perturbation scenarios.

Line 351: Some text discussions should be added for the median values as they are part of Tables 3-5, Figures 2-3.

Line 342: "As DE1 and US3 use the same SMOKE emissions and CTM" - but they appear to use different CTMs (i.e., COSMO-CLM/CMAQ for DE1, WRF/CAMx for US3)?

Table 2: There are four exposure-response coefficients for RAD in the table. How were they used in this study?

Table 4: Definition of "PD" is missing. Units should be added, as they differ across different health impacts. The same applies to Tables S2-S4. Also, please check the units for BUC and BUA in Table 2.

Figure 1: Units should be added.

Figure 2: "Days" should be replaced by "Months". "$O_3$", "$SO_2$", "$PM_{2.5}$" – please use lower case for the number.

Figure 4: Units should be added in Figures 4A and 4B. It is not clear what was shown in Figures 4C and 4D. This needs to be explained in the figure caption.

---

## Referee Comment (RC3) · Anonymous Referee #3 · 2 Dec 2017

This paper conducted a multi-model ensemble of regional models to simulate air quality over Europe and the U.S. for 2010 in the frame of AQMEII3. It estimated the impact of air pollution ($PM_{2.5}$, $O_3$, CO and $SO_2$) on human health and the associated external costs over the two continents using a common health assessment approach. Furthermore, the authors also conducted several emission perturbation scenarios to investigate the domestic and foreign contributions to the related health impacts. Overall this study is interesting and the manuscript is relatively logically organized. However, several parts of the manuscript need to be clarified and figures should be plotted more clearly. Following general and specific comments should be addressed before the publication in Atmospheric Chemistry and Physics.

**General comments:**

- The multi-model ensemble approach is widely used, especially in forecast studies in which observations are not available to evaluate the performance of individual models. Here the authors use multi-model ensemble results to investigate the air pollution levels in 2010, where sufficient measurements are available over Europe and the U.S. Therefore, the authors should show that the ensemble results are better than any individual models. As shown in Table 3 and Table 6, the RSME of multi-model ensemble results ($MM_m$ and $MM_{opt}$) are even larger than those of individual model results. Since the equations and datasets used to calculate these statistics in Tables 3 and 6 are unclear, it is difficult to judge the performance of the ensemble results. Particularly, the DE1_SMOKE simulation over the U.S. significantly underestimates $SO_2$, CO, and $PM_{2.5}$ (even up to a factor of three) comparing with the observations, which means that this result has systematic bias. This model should be removed from the ensemble, but I am not sure how it is being treated in the optimal-reduced multi-model ensembles. More description and explanations are needed here.

-This study mainly focuses on estimating the air pollution related health impacts, where annual mean concentrations of CO, $SO_2$ and $PM_{2.5}$ and yearly sum of daily maximum 8-hour $O_3$ running average over 35 ppb are used in the EVA system. The model evaluation in Section 3.1 should focus more on the spatial distribution of these models' performance, rather than on the average over the whole region. Furthermore, the authors should provide more necessary information for model evaluation, e.g., sources of observations, equations used to calculate the statistics, etc.

-From the model evaluation, it shows that results from different models have large divergence. This should be caused by many factors, like emissions, transport, chemistry, dry/wet removals. I would suggest the authors provide more information about the mechanisms/parameterizations used for each model in the supporting materials.

-In this study, the intercontinental impacts are investigated using the 20 % emission reduction scenarios applied over the source regions. In their model experiments, a

global model was used to provide chemical boundary conditions for all participating regional models. To my knowledge, the long-range transport of air pollutants is controlled by many complicated factors, which may lead to much larger uncertainties over the long-distance path than the source region. I am not sure that using a single model to represent the long-range transport is a proper way for an ensemble analysis. Therefore, the authors should provide more information regarding the evaluation of the global model.

- Figure quality is low and needs improvement, especially for Figures 1 and 4. The authors should make font-size, colorbar size, subtitles, units, and plot captions consistent. See specific comments below.

**Specific comments:**

Lines 102-116: This paragraph introduced a number of previous works quantifying air pollution-related health impacts due to intercontinental transport. However, the results of those studies showed inconsistent relative importance of domestic versus foreign emissions. Please comment on this.

Lines 250-251: " … previous AQMEII-related works" need to show some references here.

Lines 254-255: The authors should briefly introduce the sources and features of these observation data used in this study.

Lines 329-330: The authors should describe in detail how the observed and simulated monthly time series in Figures 2 and 3 are obtained. For example, whether or not the observed and simulated results averaged over the whole continental regions are sampled with identical time and locations.

Lines 390-391: "…the numbers of cases are strongly correlated to the population density…", please refers to Figure 1 for comparison.

Table 6: Why not use the same units for Europe and North America?

Figure 1: Please clarify which continent the left/right panel refers to in the caption. The unit of population density also needs to be provided. More detailed terrestrial boundaries are recommended to distinguish countries or states. Furthermore, I recommend using the same scale for the two panels to have a better comparison.

Figure 4: besides the same comments for Figure 1, figure quality needs to be improved significantly. The authors should be consistent in making the plots. For example, the top two plots have subtitles while the bottom ones don't. The font-size and colorbar

size of these panels are different. The units are missed in the top two panels. The colorbar of plot (d) even overlaps the coordinate. Additionally, the caption does not provide all necessary information to understand this figure.

---

## Referee Comment (RC4) · Anonymous Referee #4 · 4 Dec 2017

Summary comments

This manuscript is an ambitious effort to simulate air quality changes and estimate health impacts using an ensemble of models. The results clearly reflect a substantial effort on the part of the authors. I have three primary concerns: (1) the health impact assessment is insufficiently documented. In particular, the manuscript does not clearly describe the procedure for selecting and applying health endpoints to quantify or the source of the baseline incidence rates in the U.S. and Europe. (2) Reasonable people can disagree as to whether it's appropriate to quantify the economic value of years

of life lost. However, the manuscript does not attempt to provide a rationale for this choice. (3) Finally, the authors should indicate whether each of the air quality and health impact models used have been peer reviewed and whether the source code is publicly available.

Detailed comments

Line 46: Is this correct? The outdoor air pollution portion of the Global Burden of Disease studies have applied a consistent modelling framework to predict ambient concentrations of common air pollutants, and quantify the number of premature deaths attributable to outdoor fine particles and ground-level ozone. Other studies, including Anenberg et al. (2010, 2014) quantify global ozone and PM-attributable deaths due to anthropogenic emissions. Line 50: Anthropgenic and non-anthropogenic? Line 53: Did you estimate impacts down to some background concentration, or to zero? Lines 52-65: Here and elsewhere it would be helpful to distinguish between the air quality modeling portion of the ensemble and the health impact estimation portion of the ensemble. Lines 66-77: Are these a sum of the PM2.5 and ozone-related premature deaths? Line 85: What does "scale dependent challenge" mean in this context? Line 93: Suggest updating with most current GBD published value. Lines 104-109: These two statements are difficult to reconcile. Line 150: This isn't quite right. That paper estimated a total of between 130k and 350k PM & O3 related deaths. Note also that this paper quantified impacts from anthropogenic emissions alone. Line 155: Suggest rephrasing for clarity. Lines 197-202: I had a hard time following these statements. In particular, I could not understand what exactly you did to minimize error and what redundancy you're referring to. Line 291: How does this ozone metric correspond to the exposure metrics specified in each epidemiological study? Line 292: Here (or elsewhere) it would be useful to provide the rationale for selecting these health endpoints. Citing back to WHO or US EPA documents or other systematic reviews would be helpful. Line 297: It's really difficult to understand why YOLL are being divided by 10.6. Why not simply quantify counts of excess cases in the EVA tool? Line 300: the

selection of c-r functions greatly influences the health impact assessment, and so I'd recommend including this information directly in the manuscript rather than citing back to another paper. Likewise, what is the source of the baseline death and morbidity rates? At what spatial scale were these data available? Lines 303-321: I'd suggest providing a clearer rationale for valuing years of life lost rather than counts of excess death. Line 314: Please provide a citation to support this claim. Line 316: Did you consider adjusting the WTP to account for changes in income over time (i.e. income elasticity)? Line 320: Why adjust the WTP using a PPP when you can just apply a U.S. specific value? Line 394-402: Please report the currency year. Line 418: Did you consider reporting population-normalized results (e.g. deaths per 100k)? Line 434: Can you clarify what a health impact index is? Table 2: The nomenclature is a little misleading. In a health impact function, effect coefficients are exponentiated and multiplied against an air quality change and then against baseline incidence rate and the population exposed. However, the effect coefficient is written as "x cases/ugm3". This is not correct. Table 2: Several of the endpoints list multiple studies. Were these pooled in some way? Tables 3-4: Please include 95% confidence intervals

---

## Referee Comment (RC5) · Anonymous Referee #5 · 7 Dec 2017

This study addresses the impacts of air pollution on health and their economic evaluation in Europe and the US. The study has been done within the AQMEII3 action. There are several aspects: a CTM ensemble, health impact assessment and economic evaluation. Numerous research teams in Europe and the US have coauthored the manuscript.

The study contains new and important results and definitely deserves to be published. However, in my view the presentation of methods and results in the manuscript should be improved, as detailed below.

General comments

First, the description of the health impact assessments and the economic impacts should be more detailed, and include especially all the assumptions and choices made in making the computations and assessments. There are numerous alternative choices that you will need to make for e.g. economic evaluations; some of these have been properly described and discussed, whereas some have not been described. Reviewer number 1 has already detailed this issue.

Second, there are also gaps in the description of the individual CTM's and, the constructed ensemble and the evaluation of the models and the ensemble. In particular, there is very little discussion on how the non-anthropogenic emission sources have been included; as these constitute a substantial part of the total PM mass, these should also be described. There should be also discussion on the main limitations of the CTM's and the emission inventories used, what are their main uncertainties and the most poorly known parts of modelling. Details on this issue are in 'detailed comments'.

Regarding model evaluation, the manuscript should specify which networks of stations were used, how many stations were considered within each domain, and what were their site classifications. Large PM deficits were found for some models. The manuscript should therefore discuss the most probably reasons for these underpredictions: were these caused by deficiencies of the used CTM's, missing emissions or both, or/and some other reason.

Regarding the presentation of the results, there are a lot of large tables, but in my view too little synthesis and graphical illustration of the main results and findings. I would recommend to move some of the large tables an annex or to supplementary materials for better readability, and some summary figures could be added instead, to highlight the main insights, findings and conclusions. Regarding the section 'materials and methods', I recommend to use the traditional sections for a better readability, e.g.,

first Evaluation of emissions, then Atmospheric dispersion modelling, the construction of ensembles, Health impact assessment and finally economic parts. The current subtitles list one project and one model.

Detailed comments

Abstract.

Lines. 52-53. This is one of the main results of the study, so it should be presented clearly. This study addresses models for (i) emissions, (ii) dispersion, (iii) health assessment and (iv) economic evaluation. The term 'model' should therefore be used carefully and specified as necessary, throughout the manuscript. This sentence probably refers to CTM's but not health models (or emission models). It is therefore variation due to the differences of CTM's. However, the computed health impacts can also vary a lot depending on which health assessment model would be used, and which health assessment assumptions would be selected. In this study, the authors have addressed the variability due to CTM's but not that of the health assessment modelling, although the latter uncertainty is commonly much larger. Please clarify and write more clearly and accurately what is meant.

Lines 54-55. These results could be also presented per capita; this would better illustrate better the differences of the two selected domains. The PM concentration levels and the distributions of population of the two domains could also be quantitatively compared. 'In agreement', specify quantitatively, e.g., within what percentage.

Line 68. Write the acronym in full.

Line 71. 'global anthropogenic emissions' – specified for which pollutant species ?

Line 72. 'emissions foreign emission' – correct sentence

Lines 75-77. 'foreign sources make a minor contributing . . .'. This is too general. Whether the sources in a specified domain contribute more or less to health within that domain depends on a lot of factors, such as e.g., population densities in the considered

areas, how large the considered two areas are, which pollutants are considered, etc. This statement is therefore correct for some cases, and not correct for some others. Please rewrite the statement more accurately.

Introduction

Lines 107-109, and lines 114-117. Same comment as above. Whether these statements are true, depends on various factors – the relevant factors therefore need to be specified.

Lines 134-136. When presenting cost values, it is proper to state also for which year this has been evaluated.

Line 168. '. . . seen . . . ' - correct the English language.

Lines 200-202. Using a so-called optimal ensemble is fine, but as far as I know, it does not guarantee that there is e.g. no redundancy or recursiveness of models. Practically in all cases, a collection of CTM's will have some very similar treatments; using an 'optimal' ensemble will probably reduce their effect, and that is OK, but it does not altogether remove these effects.

Materials and methods

Line 218. Should read 'emission information'. There are also several other input datasets, obviously. Report also the modelling of sea salt, desert dust, biogenic emissions, wild-land fires, etc. Add some discussion on what were the main limitations, uncertainties and gaps of modelling of the CTM's used.

Results

What were the networks of stations used in Europe and the US; these should be described. How many stations were considered ? What were the classifications of stations – were all of these classified as regional or global background ?

Conclusions

Line 562. This statement may be true, but it should be supported by quantitative evidence: were there model runs to quantify this effect, and how large was it in e.g. per cents of predicted concentrations ? Alternatively, if not confirmed, this statement could be removed.

Lines 533-538. The underestimation of PM mass is a key uncertainty. There should therefore be some accurate assessment on the reasons resulting to this uncertainty. For instance, 'natural emissions' are mentioned, but it is not stated in the text which of these were included, which were neglected, and which possible omission or underestimation could probably have the largest effect. Please add some discussion of the most probable causes of the under-prediction.

---

## Author Comment (AC1) · 24 Jan 2018

We thank the reviewer for the constructive comments. We have tried to address all the points raised in the review.

*Comment: Lines 84-96 should be updated with the most recent GBD 2016 numbers*

Response: The numbers are updated (Lines 87-89).

*Comment: Lines 118-153 could use some organization. This section is basically just listing results from individual studies without synthesizing them or connecting them to the present study. It's not clear as written by this section is there.*

Response: We have now extended this section (Lines 120-132).

*Comment: Line 188-190 states that this is the first study to use a common approach for health impact assessment across US and Europe, but the HTAP ozone and PM2.5 health impact assessments referenced earlier used a similar approach. Perhaps the authors are referring only to the economic valuation portion? If so, I'm still not sure this is the first study to do that since there are now several (perhaps many) global health impact and valuation studies that use a common approach for all countries/regions, including US and Europe.*

Response: The economic valuation was not included in the GBD assessment and others. OECD has published a global assessment with economic valuation, but without a consistent atmospheric modelling framework.

*Comment: Lines 296-298: given that this paper's focus is on the health impacts, and not the modeling, there should be much more detail given here about the health impact methods in addition to, or instead of, the modeling detail, which can be found in other places and referenced. The health methods quickly summarized here diverge from the methods used by the Global Burden of Disease, U.S. EPA, and many recently published papers. So this needs to be explained, expanded, and justified quite a bit more. As stated, summing ozone deaths with PM2.5 YOLL doesn't make logical sense, as one is cases and one is years, and what is being divided by 10.6 and why? The CAFÉ reference is 12 years old, and air pollution epidemiology and health impact assessment has advanced quite a bit since then. For ozone, there are now studies showing effects of long-term exposure on mortality, just like for PM, so why are only short-term ozone impacts calculated?*

Response: EVA methodology is now extended (Lines 326-420). The selected health end-points are fairly conventional and aligned to the impact assessments that have been done for the European Commission and the European Environment Agency (EEA) up to 2013; they have been richly documented elsewhere. It was not the purpose here to develop a novel health impact assessment, or to compare in detail with GBD or US-EPA, but rather to explore its implications across the two continents.

*Comment: Lines 299-302: The ERFs listed in Table 2 are quite a bit out of date, particularly for the U.S. studies. Most of these are 20 years old. There have been many studies now reporting updated ozone and PM2.5 risk estimates for the American Cancer Society cohort which can be used. And these are not necessarily consistent magnitudes compared with the old studies.*

Response: These ERFs are consistent with the functions used by the EEA and conservative as they are updated only if recommended by the WHO even though there are recent studies providing updated functions. This is now added to the manuscript. A new version of the model is currently under development with more updated ERFs, additional species such as $NO_2$, chronic $O_3$-related mortality, and a breakdown of the aerosol components.

*Comment: Table 2 needs concentration metrics to which each ERF applies. Section 2.2 should state which concentration metrics were drawn from the models (annual average, annual average of 8-hr daily max, etc.) used which each ERF. I see now these are indicated starting in line 376, but not explained, and should be in section 2.2.*

Response: Table 2 includes which pollutants are used for each health impact. The section is also extended now to include more specifically what metric are used on what temporal resolution (Lines 358-360), following: "EVA calculates and uses the annual mean concentrations of CO, $SO_2$ and $PM_{2.5}$, while for $O_3$, it uses the SOMO35 metric that is defined as the yearly sum of the daily maximum of 8-hour running average over 35 ppb, following WHO (2013) and EEA (2017)."

*Comment: Section 2.2 should also give some equations used to calculate health impacts. It's difficult to understand what was done and impossible to judge whether it's technically sound.*

Response: We have now extended this section and it is now clearer on the implementation of the model (Lines 326-420).

*Comment: Section 2.2 were the exposure response functions applied in a linear equation or some other functional form (e.g. log-linear)? This is important for the perturbation simulations because you are reducing pollution at the high end, where the shape of the curve can have a big impact on the magnitude of health benefits estimated.*

Response: We have now added the following sentence (Lines 353-355): "EVA uses ERFs that are modelled as a linear function, which is a reasonable approximation as showed in several studies (e.g. Pope et al., 2000; the joint World Health Organization/UNECE Task Force on Health (EU, 2004; Watkiss et al., 2005))."

*Comment: Section 2.2 should also indicate the source of baseline disease rates to calculate health impacts.*

Response: the EVA model applies universal baseline rates from Statistics Denmark, therefore not country-specific, which is a simplification, although aligned to the Eurozone countries.

*Comment: Section 2.2 did you first estimate health impacts from each individual model and then average, or first average the concentrations across models and then estimate health impacts?*

Response: We have now extended the section (Lines 288-294). All modeling groups interpolate their model outputs on a common 0.25°×0.25° resolution AQMEII grid predefined for Europe (30°W - 60°E, 25°N - 70°N) and North America (130°W - 59.5°W, 23.5°N - 58.5°N). All the analyses performed in the present study use the pollutant concentrations on these final grids. Health impacts are first calculated for each individual model and then the ensemble mean, median and standard deviation are calculated for each health impact. In order to be able to estimate an uncertainty in the health impacts calculations, none of the models were removed from the ensemble.

*Comment: Section 2.2 what spatial resolution was used to estimate health impacts? Part of the problem with previous studies of PM long-range transport is that the grid resolution was too coarse to adequately capture health benefits from reducing local PM. Spatial scale is important.*

Response: We have now extended the section (Lines 288-294). All modeling groups interpolate their model outputs on a common 0.25°×0.25° resolution AQMEII grid predefined for Europe (30°W - 60°E, 25°N - 70°N) and North America (130°W - 59.5°W, 23.5°N - 58.5°N). All the analyses performed in the present study use the pollutant concentrations on these final grids. Health impacts are first calculated for each individual model and then the ensemble mean, median and standard deviation are calculated for each health impact. In order to be able to estimate an uncertainty in the health impacts calculations, none of the models were removed from the ensemble.

*Comment: Section 3.2 are the plus/minus numbers given with all the results the range of health impacts calculated with individual models? How was uncertainty in the exposure response function accounted for?*

Response: We have now added the following (Lines 291-294). Health impacts are first calculated for each individual model and then the ensemble mean, median and standard deviation are calculated for each health impact.

*Comment: Line 413 appears to be missing a 0 in the HTAP2 result*

Response: We have now corrected this.

*Comment: Line 421 what is meant by "by construction"?*

Response: We have removed this phrase.

*Comment: There are many references to the Liang (in preparation) study, but since this study is not yet available the usefulness of these comparisons is limited. It is often used as justification that the present study was done right, since the numbers match up. But there is not currently enough information from either study to judge that.*

Response: We have added more comparisons with other published studies (Lines 563-566; 617-620).

*Comment: There are many tables with numbers for health impacts that are difficult to digest. Suggest replacing some of these with figures to highlight the most salient points.*

Response: We have moved some of the tables (Table 3 and Table 4) in the supplement and kept the ensemble mean results together with the optimal ensemble results from old Table 7 to the new Table 3. However, we believe that these numbers should be explicitly presented in the manuscript as particularly the morbidity calculations are for the first time calculated for both continents and transferring them into figures would lose the details.

---

## Author Comment (AC2) · 24 Jan 2018

We thank the reviewer for the comments. We have responded to all the points raised in the review.

General comments:

*Comment: The Abstract is a bit too long. I encourage the authors to shorten their abstract to make it concise and informative. In addition, the authors should be more careful about the units. Many units in the tables and figures are missing or unclear and should be added.*

Response: The abstract is now shortened, however more details are added based on comments from the other reviewers.

*Comment: Although the description of the methods is comprehensive, additional description is needed. As the ensemble-contributing members as well as the gridded population density data have different spatial resolutions(see Table 1), the combining methods for those data should be added. Also, what is the spatial resolution of the multi-model ensemble mean (MMm) and the optimal reduced ensemble mean (MMopt) (Fig. 4)?*

Response: We have now extended the section (Lines 288-294). All modeling groups interpolate their model outputs on a common 0.25°×0.25° resolution AQMEII grid predefined for Europe (30°W - 60°E, 25°N - 70°N) and North America (130°W - 59.5°W, 23.5°N - 58.5°N). All the analyses performed in the present study use the pollutant concentrations on these final grids. Health impacts are first calculated for each individual model and then the ensemble mean, median and standard deviation are calculated for each health impact. In order to be able to estimate an uncertainty in the health impacts calculations, none of the models were removed from the ensemble.

Specific comments:

*Comment: Line 72: "North American emissions foreign emissions"-delete "foreign emissions".*

Response: We have corrected the sentence.

*Comment: Line 224-225: "a number of emission perturbation scenarios have been simulated (Table 1)"–there is noEAS emission perturbation scenario for the European domain, and no EUR emission perturbation scenario for the North American domain. Please explain the design of the perturbation scenarios.*

Response: We have now extended the section for emission perturbation scenarios (Lines 265-286).

*Comment: Line 351: Some text discussions should be added for the median values as they are part of Tables 3-5, Figures 2-3.*

Response: We have now added results on the median values in the manuscript (Lines 482-484; 511-515; 547-550; 611-614).

*Comment: Line 342: "AsDE1 and US3 use the same SMOKE emissions and CTM"-but they appear to use different CTMs (i.e., COSMO-CLM/CMAQfor DE1, WRF/CAMxfor US3)?*

Response: US3 also uses the CMAQ model. This is now corrected in the text and tables.

*Comment: Table 2: There are four exposure-response coefficients for RAD in the table. How were they used in this study?*

Response: The ERF for RAD is actually calculated as an equation. The first term of the equation is the global ERF, and the subsequent three components represent deductions of RADs as related to the three hospitalizations (to avoid double counting of the days involved). The second term represents the respiratory admission due to PM, the fourth term represents cerebrovascular admissions due to PM and the third term is calculated only for the adults above 65.

*Comment: Table 4: Definition of "PD" is missing. Units should be added, as they differ across different health impacts. The same applies to TablesS2-S4. Also, please check the units for BUC and BUA in Table 2.*

Response: Definition of PD is now added to the captions. All units for health impacts are provided in Table as either number of cases or number of days.

*Comment: Figure 1: Units should be added.*

Response: The unit is added in the figure caption.

*Comment: Figure 2: "Days" should be replaced by "Months"."O3", "SO2", "PM2.5"–please use lower case for the number.*

Response: We have corrected the figure caption.

*Comment: Figure 4: Units should be added in Figures 4A and 4B. It is not clear what was shown in Figures4C and 4D.This needs to be explained in the figure caption.*

Response: We have modified the figure caption.

---

## Author Comment (AC3) · 24 Jan 2018

We thank the reviewer for the review. We have tried to implement all the comments and corrections in the new manuscript.

**General comments:**

*Comment: The multi-model ensemble approach is widely used, especially in forecast studies in which observations are not available to evaluate the performance of individual models. Here the authors use multi-model ensemble results to investigate the air pollution levels in 2010, where sufficient measurements are available over Europe and the U.S. Therefore, the authors should show that the ensemble results are better than any individual models. As shown in Table 3 and Table 6, the RSME of multi-model ensemble results (MMm and MMopt) are even larger than those of individual model results. Since the equations and datasets used to calculate these statistics in Tables 3 and 6 are unclear, it is difficult to judge the performance of the ensemble results. Particularly, the DE1_SMOKE simulation over the U.S. significantly underestimates SO2, CO, and PM2.5 (even up to a factor of three) comparing with the observations, which means that this result has systematic bias. This model should be removed from the ensemble, but I am not sure how it is being treated in the optimal-reduced multi-model ensembles. More description and explanations are needed here.*

Response: We have now extended the description and the discussion on mean and median multi-model results (Lines 482-484; 511-515; 547-550; 611-614). In order to be able to estimate an uncertainty in the health impacts calculations using concentration inputs from different models, none of the models were removed from the ensemble. It is true that the multi model mean results do not outscore all individual models and that is why we present both individual model results and multi-model ensemble results in the manuscript.

*Comment: This study mainly focuses on estimating the air pollution related health impacts, where annual mean concentrations of CO, SO2 and PM2.5 and yearly sum of daily maximum 8-hour O3 running average over 35 ppb are used in the EVA system. The model evaluation in Section 3.1 should focus more on the spatial distribution of these models' performance, rather than on the average over the whole region. Furthermore, the authors should provide more necessary information for model evaluation, e.g., sources of observations, equations used to calculate the statistics, etc.*

Response: We have now added spatial model performance based on the bias (Figures 4 and 5) and included the relevant discussion (Lines 485-499; 516-528).

*Comment: From the model evaluation, it shows that results from different models have large divergence. This should be caused by many factors, like emissions, transport, chemistry, dry/wet removals. I would suggest the authors provide more information about the mechanisms/parameterizations used for each model in the supporting materials.*

Response: We have now added more details in Table 1 and model system descriptions in the supplementary material adopted from Solazzo et al., 2017.

*Comment: In this study, the intercontinental impacts are investigated using the 20 % emission reduction scenarios applied over the source regions. In their model experiments, a global model was used to provide chemical boundary conditions for all participating regional models. To my knowledge, the long-range transport of air pollutants is controlled by many complicated factors, which may lead to much larger uncertainties over the long-distance path than the source region. I am not sure that using a single model to represent the long-range transport is a proper way for an ensemble analysis. Therefore, the authors should provide more information regarding the evaluation of the global model.*

Response: We have used one global model to produce the boundary conditions to the regional CTMs in order to limit the uncertainty in the multi-model ensemble. The evaluation of the global is not the aim of this study s it is a common input to all the regional models. C-IFS model has been extensively evaluated elsewhere (e.g. Flemming et al. (2015 and 2017), and in particular for the North America in Hogrefe et al. (2017) and Huang et al. (2017).

*Comment: Figure quality is low and needs improvement, especially for Figures 1 and 4. The authors should make font-size, colorbar size, subtitles, units, and plot captions consistent. See specific comments below.*

Response: We have now improved the figures.

**Specific comments:**

*Comment: Lines 102-116: This paragraph introduced a number of previous works quantifying air pollution-related health impacts due to intercontinental transport. However, the results of those studies showed inconsistent relative importance of domestic versus foreign emissions. Please comment on this.*

Response: These studies uses different sets of global models on different spatial resolutions. However results were consistent in terms of the contribution of local vs. non-local sources on the impacts of pollution.

*Comment: Lines 250-251: "… previous AQMEII-related works" need to show some references here.*

Response: These references are already listed in Lines 301-302.

*Comment: Lines 254-255: The authors should briefly introduce the sources and features of these observation data used in this study.*

Response: We have now added information on the source of the observations (Lines 250-259): "The observational data used in this study are the same as the dataset used in second phase of AQMEII (Im et al., 2015a, b). Surface observations are provided in the Ensmeble system (http://ensemble2.jrc.ec.europa.eu/public/) that is hosted at the Joint Research Centre (JRC). Observational data were originally derived from the surface air quality monitoring networks operating in EU and NA. In EU, surface data were provided by the European Monitoring and Evaluation Programme (EMEP, 2003; http://www.emep.int/) and the European Air Quality Database (AirBase; http://acm.eionet.europa.eu/databases/airbase/). In NA observational data were obtained from the NAtChem (Canadian National Atmospheric Chemistry) database and from the Analysis Facility operated by Environment Canada (http://www.ec.gc.ca/natchem/)."

*Comment: Lines 329-330: The authors should describe in detail how the observed and simulated monthly time series in Figures 2 and 3 are obtained. For example, whether or not the observed and simulated results averaged over the whole continental regions are sampled with identical time and locations.*

Response: We have now added the following (Lines 244-250): "The models' performance on simulating the surface concentrations of the health-related pollutants were evaluated using Pearson's Correlation ($r$), normalized mean bias ($NMB$), normalized mean gross error ($NMGE$) and root mean square error ($RMSE$) to compare the modelled and observed hourly pollutant concentrations over surface measurement stations in the simulation domains. The hourly modelled vs. observed pairs are averaged and compared on a monthly basis. The modelled hourly concentrations were first filtered based on observation availability before the averaging has been performed."

*Comment: Lines 390-391: "…the numbers of cases are strongly correlated to the population density…", please refers to Figure 1 for comparison.*

Response: We have now referred to Fig. 1 (Line 590).

*Comment: Table 6: Why not use the same units for Europe and North America?*

Response: We have now corrected the captions. Units are consistent over the two domains.

*Comment: Figure 1: Please clarify which continent the left/right panel refers to in the caption. The unit of population density also needs to be provided. More detailed terrestrial boundaries are recommended to distinguish countries or states. Furthermore, I recommend using the same scale for the two panels to have a better comparison.*

Response We have now updated Fig. 1.

*Comment: Figure 4: besides the same comments for Figure 1, figure quality needs to be improved significantly. The authors should be consistent in making the plots. For example, the top two plots have subtitles while the bottom ones don't. The font-size and colorbar size of these panels are different. The units are missed in the top two panels. The colorbar of plot (d) even overlaps the coordinate. Additionally, the caption does not provide all necessary information to understand this figure.*

Response: We have now updated Fig. 4 (now Fig. 6).

---

## Author Comment (AC4) · 24 Jan 2018

We thank the reviewer for the constructive comments. We have responded to all the comments in the new version of the manuscript.

Summary comments

*Comment: This manuscript is an ambitious effort to simulate air quality changes and estimate health impacts using an ensemble of models. The results clearly reflect a substantial effort on the part of the authors. I have three primary concerns:*

*(1) the health impact assessment is insufficiently documented. In particular, the manuscript does not clearly describe the procedure for selecting and applying health endpoints to quantify or the source of the baseline incidence rates in the U.S. and Europe.*

Response: The selected health end-points are fairly conventional and aligned to the impact assessments that have been done for the European Commission and the European Environment Agency (EEA) up to 2013; they have been richly documented elsewhere. It was not the purpose here to develop a novel health impact assessment, but rather to explore its implications across the two continents. A new generation of health impact assessments are expected to make reference to the meanwhile established WHO HRAPIE consensus guidelines.

*(2) Reasonable people can disagree as to whether it's appropriate to quantify the economic value of years of life lost. However, the manuscript does not attempt to provide a rationale for this choice.*

Response: This is a fairly crucial aspect of mortality impacts, which EU and USA simply approaches differently – we here adhere to the European approach, the main advocate of which was Ari Rabl (Rabl, Spadaro and Holland, 2014). See further below.

*(3) Finally, the authors should indicate whether each of the air quality and health impact models used have been peer reviewed and whether the source code is publicly available.*

Response: As seen in Table 1 and now in the supplementary material, there a number of CTMs used in the AQMEII exercise. Some of these CTMs are community models, such as WRF/Chem, CMAQ and CAMx, while others are not community models and being used by the main developers so that the model is not publicly available but can be shared upon collaboration. Only one health impact model has been used, using different concentration inputs from each of the CTMs. EVA system is not a community model either and developed internally by Aarhus University, but has been used upon collaboration with other institutes.

Detailed comments

*Comment: Line 46: Is this correct? The outdoor air pollution portion of the Global Burden of Disease studies have applied a consistent modelling framework to predict ambient concentrations of common air pollutants, and quantify the number of premature deaths attributable to outdoor fine particles and ground-level ozone. Other studies, including Anenberg et al. (2010, 2014) quantify global ozone and PM-attributable deaths due to anthropogenic emissions.*

Response: GBD does not provide economic estimates. Same for Anenberg et al. (2010 and 2014).

*Comment: Line 50: Anthropgenic and non-anthropogenic?*

Response: The perturbation simulations target anthrpogenic emissions. This is now added to the text (Line 49).

*Comment: Line 53: Did you estimate impacts down to some background concentration, or to zero?*

Response: EVA system uses a cut off value of 35 ppb to calculate health impacts from ozone and used to calculate the SOMO35 metric. Regarding PM2.5, no threshold is being applied, following the EEA recommendations (See Line 388-396).

*Comment: Lines 52-65: Here and elsewhere it would be helpful to distinguish between the air quality modeling portion of the ensemble and the health impact estimation portion of the ensemble.*

Response: The health impacts are calculated from each CTM individually. Therefore, the health impact ensemble includes health impacts using concentrations from the different CTMs. We have now made this more clear in the text as follows (Lines 288-294): "All modeling groups interpolate their model outputs on a common 0.25°×0.25° resolution AQMEII grid predefined for Europe (30°W - 60°E, 25°N - 70°N) and North America (130°W - 59.5°W, 23.5°N - 58.5°N). All the analyses performed in the present study use the pollutant concentrations on these final grids. Health impacts are first calculated for each individual model and then the ensemble mean, median and standard deviation are calculated for each health impact."

*Comment: Lines 66-77: Are these a sum of the PM2.5 and ozone-related premature deaths?*

Response: The numbers reflect the total premature death. The text now reads (Lines 63-71): "A total of 54 000 and 27 500 premature deaths can be avoided by a 20% reduction of global anthropogenic emissions in Europe and the U.S., respectively. A 20% reduction of North American anthropogenic emissions avoids a total premature death of ~1 000 in Europe and 25 000 total premature deaths in the U.S. A 20% decrease of anthropogenic emissions within the European source region avoids a total premature death of 47 000 in Europe. Reducing the East Asian anthropogenic emissions by 20% avoids ~2000 total premature deaths in the U.S. These results show that the domestic anthropogenic emissions make the largest impacts on premature death on a

continental scale, while foreign sources make a minor contributing to adverse impacts of air pollution."

*Comment: Line 85: What does "scale dependent challenge" mean in this context?*

Response: We have modified the sentence to be more clear (Line 79-81): "Air pollution is a transboundary phenomenon with global, regional, national and local sources, leading to large differences in the geographical distribution of human exposure."

*Comment: Line 93: Suggest updating with most current GBD published value. Lines 104-109: These two statements are difficult to reconcile.*

Response: This part has been modified with newer numbers and for better readability (Lines 87-89): "The Global Burden of Disease Study 2015  estimated 254 000 $O_3$-related and 4.2 million anthropogenic $PM_{2.5}$-related premature deaths per year (Cohen et al., 2017)."

*Comment: Line 150: This isn't quite right. That paper estimated a total of between 130k and 350k PM & O3 related deaths. Note also that this paper quantified impacts from anthropogenic emissions alone.*

Response: We have now corrected the sentence as (Line 153-155): "Fann et al. (2012) calculated 130,000 - 350,000 premature deaths associated with $O_3$ and $PM_{2.5}$from the anthropogenic sources in the U.S. for the year 2005."

*Comment: Line 155: Suggest rephrasing for clarity.*

Response: We have changed the sentence as (Line XXX): "Observations have spatial limitations particularly when assessments are needed for large regions."

*Comment: Lines 197-202: I had a hard time following these statements. In particular, I could not understand what exactly you did to minimize error and what redundancy you're referring to.*

Response: We have now rephrased this part as follows (Lines 202-205): "Finally, following the conclusions of Solazzo and Galmarini (2015), the health impacts have been calculated using an optimal ensemble of models, determined by error minimization. This approach can assess the health impacts with reduced model bias, which we can then compare with the classically derived estimates based on model averaging. "

*Comment: Line 291: How does this ozone metric correspond to the exposure metrics specified in each epidemiological study?*

Response: SOMO35 metric is recommended by the EEA and also recommended in the latest WHO report reviewing the different ERFs. We have rephrased this part as follows (Line 358-360): "EVA calculates and uses the annual mean concentrations of CO, $SO_2$ and $PM_{2.5}$, while for $O_3$, it uses the SOMO35 metric that is defined as the yearly sum of the daily maximum of 8-hour running average over 35 ppb, following WHO (2013) and EEA (2017)."

*Comment: Line 292: Here (or elsewhere) it would be useful to provide the rationale for selecting these health endpoints. Citing back to WHO or US EPA documents or other systematic reviews would be helpful.*

Response: We have now refereed to EEA and WHO reports in several parts of the manuscript (Lines XXX).

*Comment: Line 297: It's really difficult to understand why YOLL are being divided by 10.6. Why not simply quantify counts of excess cases in the EVA tool?*

Response: see comment to lines 303-321

*Comment: Line 300: the selection of c-r functions greatly influences the health impact assessment, and so I'd recommend including this information directly in the manuscript rather than citing back to another paper. Likewise, what is the source of the baseline death and morbidity rates? At what spatial scale were these data available?*

Response: We have not extended the section describing EVA substantially (Lines 326-464).

*Comment: Lines 303-321: I'd suggest providing a clearer rationale for valuing years of life lost rather than counts of excess death.*

Response: government agencies in Europe, including the European Commission, apply a methodology for costing of air pollution that is based on accounting for lost life years, rather than for entire statistical lives as is customary in USA. Whereas the average traffic victim, for instance, is mid-aged and likely to lose about 35-40 years of life expectancy, pollution victims are believed to suffer significantly smaller losses of years (EAHEAP, 1999:64; Friedrich and Bickel, 2001). To avoid overstating the benefits of air pollution control, these are treated as proportional to the number of life years lost.

The average loss of lifeyears per victim has previously been assessed to 10.6 (calculation method explained in Andersen 2017).

*Comment: Line 314: Please provide a citation to support this claim.*

Response: OECD, 2016 reference is now added to the text (Line 440)

*Comment: Line 316: Did you consider adjusting the WTP to account for changes in income over time (i.e. income elasticity)?*

Response: Indeed- the costs reported are the net present costs related to mortality and morbidity, and WTP is expected to increase with increasing incomes in the future; however this future stream of WTP needs to be discounted back into net present values. It has been customary in EU studies to apply an income elasticity of 1.

*Comment: Line 320: Why adjust the WTP using a PPP when you can just apply a U.S. specific value?*

Response: We have now extended this section (Lines 448-464). Cost-benefit analysis in USA relating to air pollution proceeds from a standard approach whereby abatement measures preventing premature mortality are considered according to the number of statistical fatalities avoided, which are appreciated according to the value of statistical life (VSL) (presently USD 7.4 million). In contrast, and following recommendations from the UK working group on Economic Appraisal of the Health Effects of Air Pollution (EAHEAP, 1999), focus in EU has been on the possible changes in average life expectancy resulting from air pollution. In EU the specific number of life years lost as a result of changes in air pollution exposures are estimated based on lifetable methodology, and monetized with Value-Of-Life-Year (VOLY) unit estimates (Holland et al. 1999; Leksell and Rabl 2001). The theoretical basis is a life-time consumption model according to which the preferences for risk reduction will reflect expected utility of consumption for remaining life years (Hammitt 2007; OECD 2006:204). The much lower VSL values customary in Europe (presently €2.2 million) add decisively to the differences, as VOLY is deducted from this value. By using a common valuation framework according the EU approach we allow for direct comparisons of the monetary results. It follows from OECD recommendations (2012) to correct with PPP when doing such benefit transfer.

*Comment: Line 394-402: Please report the currency year.*

Response: The currency year is 2013 (Line 464).

*Comment: Line 418: Did you consider reporting population-normalized results (e.g. deaths per 100k)?*

Response: such a figure is embedded in the specific exposure-response function for mortality, which was derived from lifetable analysis, however providing lost life-years per 100k

*Comment: Line 434: Can you clarify what a health impact index is?*

Response: We have now rephrased this paragraph (Lines 636-643): "Results show that for the particular input (gridded air pollutant concentrations from individual model)-output (each health outcome) configuration, the $PM_{2.5}$ drives the variability of the different health impact and that at least 81% of the variation of the health impacts are explained by sole variations in the pollutants (i.e. without interactions: Table S3). Table S1 also shows that the most important contribution to the health impacts is from $PM_{2.5}$, followed by CO and $O_3$ (with much smaller influence though). The impact of perturbing $PM_{2.5}$ by a fixed fraction of its standard deviation on the health impact is roughly double compared to CO and $O_3$."

*Comment: Table 2: The nomenclature is a little misleading. In a health impact function, effect coefficients are exponentiated and multiplied against an air quality change and then against baseline incidence rate and the population exposed. However, the effect coefficient is written as "x cases/ugm3". This is not correct.*

Response: In EVA, we use linear functions for the ERFs. We have now added the following section (Lines 353-355): "EVA uses ERFs that are modelled as a linear function, which is a reasonable approximation as showed in several studies (e.g. Pope et al., 2000; the joint World Health Organization/UNECE Task Force on Health (EU, 2004; Watkiss et al., 2005))."

*Comment: Table 2: Several of the endpoints list multiple studies. Were these pooled in some way?*

Response: Each of the morbidity effects refer to one study each.

*Comment: Tables 3-4: Please include 95% confidence intervals*

Response: We have moved the big tables into the supplementary material and made a new Table 3, which summarizes the mean results from the different ensemble approaches. Along with the mean of all individual pollutant estimates (denoted as $MM_{mi}$ in the manuscript), we have now added the standard deviations. EVA model implements the ERF functions as linear equations and the 95% CI are not taken into account presently. We agree with the reviewer that it is important to provide these numbers, however the present study employs a frozen version of the model, where the aim is not focusing on further development of the model. We continue to further develop the model on many aspects and this comment will also be taken into account.

---

## Author Comment (AC5) · 24 Jan 2018

We thank the reviewer for the comments and corrections. We have now implemented all the points to further develop the manuscript.

General comments

*Comment: First, the description of the health impact assessments and the economic impacts should be more detailed, and include especially all the assumptions and choices made in making the computations and assessments. There are numerous alternative choices that you will need to make for e.g. economic evaluations; some of these have been properly described and discussed, whereas some have not been described. Reviewer number 1 has already detailed this issue.*

Response: The EVA methodology section has been substantially extended (Lines 326-464).

*Comment: Second, there are also gaps in the description of the individual CTM's and, the constructed ensemble and the evaluation of the models and the ensemble. In particular, there is very little discussion on how the non-anthropogenic emission sources have been included; as these constitute a substantial part of the total PM mass, these should also be described. There should be also discussion on the main limitations of the CTM's and the emission inventories used, what are their main uncertainties and the most poorly known parts of modelling. Details on this issue are in 'detailed comments'.*

Response: We have added more details in Table 1 and added model descriptions to the supplementary materials adopted from Solazzo et al. (2017).

*Comment: Regarding model evaluation, the manuscript should specify which networks of stations were used, how many stations were considered within each domain, and what were their site classifications. Large PM deficits were found for some models. The manuscript should therefore discuss the most probably reasons for these underpredictions: were these caused by deficiencies of the used CTM's, missing emissions or both, or/and some other reason.*

Response: We have extended the model evaluation part (Lines 485-499; 516-528).

*Comment: Regarding the presentation of the results, there are a lot of large tables, but in my view too little synthesis and graphical illustration of the main results and findings. I would recommend to move some of the large tables an annex or to supplementary materials for better readability, and some summary figures could be added instead, to highlight the main insights, findings and conclusions.*

Response: We have moved some of the tables (Table 3 and Table 4) in the supplement and kept the ensemble mean results together with the optimal ensemble results from old Table 7 to the new Table 3. However, we believe that these numbers should be explicitly presented in the manuscript

as particularly the morbidity calculations are for the first time calculated for both continents and transferring them into figures would lose the details.

*Comment: Regarding the section 'materials and methods', I recommend to use the traditional sections for a better readability, e.g., first Evaluation of emissions, then Atmospheric dispersion modelling, the construction of ensembles, Health impact assessment and finally economic parts. The current subtitles list one project and one model.*

Response: We have now re-structured this section following the reviewers recommendations.

Detailed comments

Abstract.

*Comment: Lines. 52-53. This is one of the main results of the study, so it should be presented clearly. This study addresses models for (i) emissions, (ii) dispersion, (iii) health assessment and (iv) economic evaluation. The term 'model' should therefore be used carefully and specified as necessary, throughout the manuscript. This sentence probably refers to CTM's but not health models (or emission models). It is therefore variation due to the differences of CTM's. However, the computed health impacts can also vary a lot depending on which health assessment model would be used, and which health assessment assumptions would be selected. In this study, the authors have addressed the variability due to CTM's but not that of the health assessment modelling, although the latter uncertainty is commonly much larger. Please clarify and write more clearly and accurately what is meant.*

Response: We have now rephrased this sentence accordingly (Lines 53-55): "Health impacts estimated by using concentration inputs from different chemistry and transport models (CTMs) to the EVA system can vary up to a factor of three in Europe (twelve models) and the United States (three models)."

*Comment: Lines 54-55. These results could be also presented per capita; this would better illustrate better the differences of the two selected domains. The PM concentration levels and the distributions of population of the two domains could also be quantitatively compared. 'In agreement', specify quantitatively, e.g., within what percentage.*

Response: We have now added normalized PD numbers (number deaths per 100 000) in the text.

*Comment: Line 68. Write the acronym in full.*

Response: We have provided the full name of the acronym (Lines 48-52): "Along with a base case simulation, additional runs were performed introducing 20% anthropogenic emission reductions

both globally and regionally in Europe, North America and East Asia, as defined by the second phase of the Task Force on Hemispheric Transport of Air Pollution (TF-HTAP2)."

*Comment: Line 71. 'global anthropogenic emissions' – specified for which pollutant species ?*

Response: Emission perturbations target anthropogenic emissions. This is now made clear in the text (Lines 63-71): "A total of 54 000 and 27 500 premature deaths can be avoided by a 20% reduction of global anthropogenic emissions in Europe and the U.S., respectively. A 20% reduction of North American anthropogenic emissions avoids a total premature death of ~1 000 in Europe and 25 000 total premature deaths in the U.S. A 20% decrease of anthropogenic emissions within the European source region avoids a total premature death of 47 000 in Europe. Reducing the East Asian anthropogenic emissions by 20% avoids ~2000 total premature deaths in the U.S. These results show that the domestic emissions make the largest impacts on premature death, while foreign sources make a minor contributing to adverse impacts of air pollution."

*Comment: Line 72. 'emissions foreign emission' – correct sentence*

Response: The sentence has been corrected (Lines 64-66).

*Comment: Lines 75-77. 'foreign sources make a minor contributing : : :'. This is too general. Whether the sources in a specified domain contribute more or less to health within that domain depends on a lot of factors, such as e.g., population densities in the considered areas, how large the considered two areas are, which pollutants are considered, etc. This statement is therefore correct for some cases, and not correct for some others. Please rewrite the statement more accurately.*

Response: We agree with the reviewer. However, the abstract is just an overall short summary of the paper so such a discussion does not fit to this section. We have now slightly rephrased the sentence as following: "These results show that the domestic anthropogenic emissions make the largest impacts on premature death on a continental scale, while foreign sources make a minor contributing to adverse impacts of air pollution."

Introduction

*Comment: Lines 107-109, and lines 114-117. Same comment as above. Whether these statements are true, depends on various factors – the relevant factors therefore need to be specified.*

Response: These studies employ global model ensembles on coarse spatial resolutions to calculate mortality due to air pollution.

*Comment: Lines 134-136. When presenting cost values, it is proper to state also for which year this has been evaluated.*

Response: The currency year is 2013 (Lines 463-464).

Comment: Line 168. ': : : seen : : : ' - correct the English language.

Response:. We have rephrased the sentence as following (Lines 171-173): "Source-receptor relationships have the advantage of reducing the computing time significantly and have therefore been extensively used in systems like GAINS (Amann et al., 2011)."

Comment: Lines 200-202. Using a so-called optimal ensemble is fine, but as far as I know, it does not guarantee that there is e.g. no redundancy or recursiveness of models. Practically in all cases, a collection of CTM's will have some very similar treatments; using an 'optimal' ensemble will probably reduce their effect, and that is OK, but it does not altogether remove these effects.

Response: We agree with the reviewer. That is why we write that we produce an optimal ensemble producing the minimum error at each time step for each pollutant, and do not say that we remove the error altogether.

Materials and methods

Comment: Line 218. Should read 'emission information'. There are also several other input datasets, obviously. Report also the modelling of sea salt, desert dust, biogenic emissions, wild-land fires, etc. Add some discussion on what were the main limitations, uncertainties and gaps of modelling of the CTM's used.

Response: We have now added more details in Table 1 and provided model descriptions in the supplementary materials, adopted from Solazzo et al. (2017).

Results

Comment: What were the networks of stations used in Europe and the US; these should be described. How many stations were considered ? What were the classifications of stations – were all of these classified as regional or global background ?

Response: We have extended the model evaluation section (Lines 244-263).

Conclusions

Comment: Line 562. This statement may be true, but it should be supported by quantitative evidence: were there model runs to quantify this effect, and how large was it in e.g. per cents of predicted concentrations ? Alternatively, if not confirmed, this statement could be removed.

Response: This is the most important gap in air pollution-related health studies and therefore needs to be investigated. Therefore, there are no studies yet that designed such an experiment. Further down, we refer to a Nordic project that works on these issues.

*Comment: Lines 533-538. The underestimation of PM mass is a key uncertainty. There should therefore be some accurate assessment on the reasons resulting to this uncertainty. For instance, 'natural emissions' are mentioned, but it is not stated in the text which of these were included, which were neglected, and which possible omission or underestimation could probably have the largest effect. Please add some discussion of the most probable causes of the under-prediction.*

Response: We have now extended this paragraph (Lines 748-754). As shown in the supplementary material, the CTMs diverge a lot on the representation of particles and their size distribution, SOA formation, as well as the inclusion of natural sources. As the anthropogenic emissions are harmonized in the models, they represent a minor uncertainty in terms of model-to-model variation. However, differences in the treatment of the temporal, vertical and chemical distributions of the particulate and volatile organic species have an influence in the model calculations and therefore lead to model-to-model variations.

---

## Author Response (AR2)

Response to Rewiever 1

*Comment: I appreciate that the authors have provided additional information on the health impact assessment methods. I continue to believe that these methods are outdated and do not reflect the state of the science in air pollution epidemiology. I do not believe this should hold up publication of the paper since they are being applied in US and Europe, where the changes in air pollution epidemiology have not changed as radically as they have in other parts of the world. I would still like to see additional information given about how different concentration-response factors and shape of the concentration-response curve might influence results and conclusions.*

Response: We thank the reviewer for his positive respond to our revised manuscript. We have now added the following part to the new revised manuscript (Lines 353-359): "Many epidemiological studies have analyzed the concentration-response relationship between ambient PM and mortality using various statistical models. In general, the shapes of the estimated curves did not differ significantly from linear. However, some studies showed non-linear relationships, being steeper at lower than at higher concentrations (e.g. Samoli et al., 2005). Therefore, linear relationships may lead to overestimated health impacts over highly polluted areas."

Response to Rewiever 2

*Comment: This paper has undergone much improvement and the authors have carefully addressed most of the comments raised by the reviewers.*

Response:  We thank the reviewer for his positive respond to our revised manuscript.

Here are a few minor suggestions:

*Comment 1) The spatial distributions of model performance are provided in the revised manuscript. However, it focused on the annual mean results for different pollutants. Considering that the daily maximum 8-hour (DM8H) O3 are used in the EVA system, we suggest using DM8H to evaluate the performance of O3 simulations.*

Response: We have now plotted the spatial distribution of the DM8H O3 bias over Europe (Fig. 4) and North America (Fig. 5) and updated the captions accordingly

*Comment 2) Fig. 1: I'd suggest changing the unit of population density to "population per km2 " since the area of grid box has not been specified in the caption.*

Response: We have updated the caption accordingly: "Population density (population per $0.25° \times 0.25°$ grid box) over a) the United States and b) Europe"

*Comment 3) Fig. 6: The caption of Fig. 6 didn't specify the density of the spatial distribution. Is the number of total premature death calculated over each grid box or other areas?*

[revised manuscript text omitted]